# ACCELERATING SINKHORN ALGORITHM WITH SPARSE NEWTON ITERATIONS

**Xun Tang**[†]**, Michael Shavlovsky**[‡]**, Holakou Rahmanian**[‡]**, Elisa Tardini**[‡]**,**
**Kiran Koshy Thekumparampil**[‡]**, Tesi Xiao**[‡]**, Lexing Ying**[†]
[†]Stanford University
[‡]Amazon
{xutang,lexing}@stanford.edu
{shavlov, holakou, ettardin, kkt, tesixiao}@amazon.com

## ABSTRACT

Computing the optimal transport distance between statistical distributions is a fundamental task in machine learning. One remarkable recent advancement is entropic regularization and the Sinkhorn algorithm, which utilizes only matrix scaling and guarantees an approximated solution with near-linear runtime. Despite the success of the Sinkhorn algorithm, its runtime may still be slow due to the potentially large number of iterations needed for convergence. To achieve possibly super-exponential convergence, we present Sinkhorn-Newton-Sparse (SNS), an extension to the Sinkhorn algorithm, by introducing early stopping for the matrix scaling steps and a second stage featuring a Newton-type subroutine. Adopting the variational viewpoint that the Sinkhorn algorithm maximizes a concave Lyapunov potential, we offer the insight that the Hessian matrix of the potential function is approximately sparse. Sparsification of the Hessian results in a fast $O(n^2)$ per-iteration complexity, the same as the Sinkhorn algorithm. In terms of total iteration count, we observe that the SNS algorithm converges orders of magnitude faster across a wide range of practical cases, including optimal transportation between empirical distributions and calculating the Wasserstein $W_1, W_2$ distance of discretized densities. The empirical performance is corroborated by a rigorous bound on the approximate sparsity of the Hessian matrix.

## 1 INTRODUCTION

Optimal transport (OT) calculates the best transportation plan from an ensemble of sources to targets (Villani et al., 2009; Linial et al., 1998; Peyré et al., 2017) and is becoming increasingly an important task in machine learning (Sandler & Lindenbaum, 2011; Jitkrittum et al., 2016; Arjovsky et al., 2017; Salimans et al., 2018; Genevay et al., 2018; Chen et al., 2020; Fatras et al., 2021). In this work, we focus on optimal transportation problem with entropic regularization:

$$\min_{P:P\mathbf{1}=r,P^\top\mathbf{1}=c} C \cdot P + \frac{1}{\eta}H(P), \tag{1}$$

where $\eta > 0$ is the entropy regularization parameter, $C \in \mathbb{R}^{n \times n}$ is the cost matrix, $c, r \in \mathbb{R}^n$ are respectively the source and target density, and $H(P) := \sum_{ij} p_{ij} \log p_{ij}$ is the entropy of $P$. The insight of using the Sinkhorn algorithm is that entropy-regularized optimal transport is equivalent to an instance of matrix scaling (Linial et al., 1998; Cuturi, 2013; Garg et al., 2020):

Find diagonal matrix $X, Y$ so that $P = X \exp(-\eta C)Y$ satisfies $P\mathbf{1} = r, P^\top\mathbf{1} = c.$[1]

The Sinkhorn algorithm (Yule, 1912) alternates between scaling the rows and columns of a matrix to a target vector, and its convergence property was first proved in Sinkhorn (1964). Theoretical results show that the Sinkhorn algorithm converges at a relatively slow rate. While the Sinkhorn algorithm satisfies exponential convergence (Franklin & Lorenz, 1989; Carlier, 2022), its best proven

---

[1]The symbol $\exp(-\eta C)$ denotes entry-wise exponential.

exponential convergence rate is often too close to one for practical use (see Section 2 for detailed discussions) and in practice it behaves more like a polynomially converging method (Altschuler et al., 2017; Ghosal & Nutz, 2022). Therefore, to reach within a sub-optimality gap $\epsilon$, the tightest bound of iteration complexity in practice tends to be $O(\text{poly}(\epsilon^{-1}))$ in early stages.

We introduce a new algorithm that greatly accelerates convergence by reducing the required iteration counts. Utilizing a well-established variational perspective for the Sinkhorn algorithm (Altschuler et al., 2017), we consider the Lyapunov potential $f \colon \mathbb{R}^n \times \mathbb{R}^n \to \mathbb{R}$ associated with the entropic optimal transport problem in equation 1:

$$f(x, y) := -\frac{1}{\eta} \sum_{ij} \exp(\eta(-c_{ij} + x_i + y_j) - 1) + \sum_i r_i x_i + \sum_j c_j y_j. \tag{2}$$

In particular, the discussion in Section 3 shows that solving equation 1 is equivalent to obtaining $(x^\star, y^\star) = \arg\max_{x,y} f(x, y)$. We emphasize that $f$ is *concave*, allowing routine convex optimization techniques to be used. Under this framework, the matrix scaling step in the Sinkhorn algorithm can be seen as an alternating maximization algorithm

$$\begin{cases} x \leftarrow \arg\max_x f(x, y), \\ y \leftarrow \arg\max_y f(x, y). \end{cases} \tag{3}$$

The formula in equation 3 provides surprisingly clear guidance and justification for our approach to accelerate the Sinkhorn algorithm. First, one can only achieve a substantially reduced iteration count by jointly optimizing $(x, y)$. Second, no first-order method can be used, as they achieve polynomial convergence at best, reaching an iteration complexity of $O(\epsilon^{-1})$ in the case of gradient descent for a sub-optimality of $\epsilon$ (Boyd & Vandenberghe, 2004), or $O(\epsilon^{-1/2})$ in the case of accelerated gradient descent (Nesterov, 1983). In conclusion, one can only hope to achieve better convergence with second-order methods, which enjoy super-exponential convergence (Boyd & Vandenberghe, 2004). The use of Newton's method for the Sinkhorn algorithm has been introduced in Brauer et al. (2017). However, even a single Newton step has an $O(n^3)$ cost, which violates the goal of having a near-linear time algorithm with $O(n^2)$ total complexity. This naturally leads to the question:

> Is there an algorithm that reaches the optimality gap $\epsilon$ with an iteration complexity of Newton's algorithm and an $O(n^2)$ per-iteration complexity of the Sinkhorn algorithm?

**Practical Newton's algorithm via Hessian sparsification**   We answer the question in the affirmative by introducing the Sinkhorn-Newton-Sparse (SNS) algorithm, which achieves fast convergence and an $O(n^2)$ per-iteration complexity. The main technical novelty is to solve the Hessian system $(\nabla^2 f)v = -\nabla f$ in Newton's algorithm efficiently with sparse approximation. In particular, we show that the Hessian matrix of the Lyapunov potential is approximately sparse in the following sense:

**Definition 1.** *(Sparsity and approximate sparsity) Let $\|\cdot\|_0$ denote the $l_0$ norm. The sparsity of a matrix $M \in \mathbb{R}^{m \times n}$ is defined by $\tau(M) := \frac{\|M\|_0}{mn}$. Furthermore, a matrix $M \in \mathbb{R}^{m \times n}$ is $(\lambda, \epsilon)$-sparse if there exists a matrix $\tilde{M}$ so that $\tau(\tilde{M}) \leq \lambda$ and $\|M - \tilde{M}\|_1 \leq \epsilon$.*

In Section 5, we prove a rigorous bound on the approximate sparsity of the Hessian matrix. We highlight our result by providing an informal version of our sparsity analysis:

> **Theorem.** *(Informal version of Theorem 1) Assume $\min_{P \colon P\mathbf{1}=r, P^\top\mathbf{1}=c} C \cdot P$ admits a unique solution. Then, if $t, \eta$ are sufficiently large, the Hessian matrix after $t$ Sinkhorn matrix scaling step is $(\frac{3}{2n}, 12n^2 \exp(-p\eta) + \frac{q}{\sqrt{t}})$-sparse for some parameter $p, q$.*

As $\nabla^2 f$ is approximately sparse, one can approximate $\nabla^2 f$ with a relaxed target sparsity $\lambda = O(1/n) > \frac{3}{2n}$. The cost for solving the sparsified linear system thus reduces to $O(\lambda n^3) = O(n^2)$.

**Contributions**   The contribution of this paper is threefold. First, we point out the sparsification technique, and the resultant SNS algorithm reduces the per-iteration cost of Newton step to $O(n^2)$.

While a related idea has been discussed with the name of the Sinkhorn-Newton algorithm in Brauer et al. (2017), the $O(n^3)$ complexity of Sinkhorn-Newton makes the algorithm prohibitively expensive to work with. Second, we give a non-asymptotic analysis, showing that one can expect sparse Hessian generically. Third, we fully adapt our sparsity argument to the case of non-unique solutions. We introduce a novel argument that shows an $O(\frac{1}{\sqrt{n}})$ sparsity. Moreover, the provided numerical analysis directly treats the case of non-unique solutions.

**Notation** For $n \in \mathbb{N}$, we denote $[n] := \{1, \ldots, n\}$. We use shorthand for several matrix operations for the sake of notational compactness. The $\cdot$ operation between matrices is defined by $C \cdot P = \sum_{i,j=1}^{n} c_{ij} p_{ij}$. For a matrix $M$, the notation $\log M$ stands for entry-wise logarithm, and similarly $\exp(M)$ denotes entry-wise exponential. The symbol $\mathbf{1}$ stands for the all-one vector in $\mathbb{R}^n$. Finally, we use the symbol $\|M\|_1$ to denote the entry-wise $l_1$ norm, i.e. $\|M\|_1 := \|\text{vec}(M)\| = \sum_{ij} |m_{ij}|$.

## 2 RELATED LITERATURE

**Convergence of Sinkhorn** We give an overview of the convergence of the Sinkhorn algorithm and discuss the iteration complexity to reach within an error tolerance $\alpha$ on the marginal KL divergence, which is defined by $\mathcal{L}(P) := \text{KL}(r\|P\mathbf{1}) + \text{KL}(c\|P^\top\mathbf{1})$. The Sinkhorn algorithm has a sub-optimality gap of $O\left(\left(1 - e^{-24\|C\|_\infty \eta}\right)^t\right)$ after $t$ steps (Carlier, 2022), which implies an iteration complexity of $O(e^{24\|C\|_\infty \eta} \log(1/\alpha))$. The analysis by Altschuler et al. (2017) proves an iteration complexity of $O(\alpha^{-1})$, and it has been refined to $O(\alpha^{-1/2})$ in special cases (Ghosal & Nutz, 2022). We refer the readers to Peyré et al. (2017); Carlier (2022) for a more comprehensive review of the convergence of the Sinkhorn algorithm. The analysis in (Mallasto et al., 2020; Weed, 2018) considers the convergence of the entropic optimal transport solution to the true transport plan, which further connects the Sinkhorn solution to the ground-truth optimal transport plan.

**Sparsification in Sinkhorn** Sparsification techniques have been extensively explored to improve the efficiency of the Sinkhorn algorithm. Sparsification has been considered during matrix scaling (Li et al., 2023), and a 2-norm or group lasso penalty function has been considered to boost sparsity in the transport plan (Blondel et al., 2018). Unlike the methods discussed, which modify the optimization task, our work retains the original formulation while introducing enhancement via sparsification inside the optimization step. Our approach uses Newton's algorithm, and the sparsity is applied to the Hessian matrix for the Lyapunov function. The numerical analysis in our work lends further support to a sparse transportation plan in the entropic case.

**Acceleration for Sinkhorn** There is considerable research interest in speeding up the runtime of the Sinkhorn algorithm. There are a few strategies related to our work, including randomized or greedy row/column scaling (Genevay et al., 2016; Altschuler et al., 2017), dynamic schedule of entropic regularization (Chen et al., 2023). These works complement our work, as both strategies can be naturally used in our Newton-type algorithm. Although both techniques can potentially boost the convergence speed of SNS, we focus on the case with fixed entropy strength and joint optimization over all variables to show that a sparse approximation to the Hessian can reach rapid convergence on its own. Another notable line of acceleration technique considers acceleration using an approximation of the kernel $K = \exp(-C\eta)$ (Deriche, 1993; Solomon et al., 2015; Bonneel et al., 2016; Altschuler et al., 2019; Scetbon & Cuturi, 2020; Huguet et al., 2023) or by exploring low-rankness of the transport plan(Scetbon et al., 2021).

**Variational methods in Entropic OT** Furthermore, a stream of research focuses on jointly optimizing the potential $f$ using accelerated first-order method (Dvurechensky et al., 2018; Thibault et al., 2021; Kemertas et al., 2023). The SNS algorithm can be extended by replacing Sinkhorn matrix scaling steps with iteration schemes based on first-order methods. As a variational interpretation of the Sinkhorn algorithm, it can also be viewed as mirror descent (Mishchenko, 2019; Léger, 2021).

**OT in Machine Learning** A rich body of literature exists on the applications of optimal transport in various machine-learning domains. Research such as Kolouri et al. (2017); Vayer et al. (2018); Genevay et al. (2019); Luise et al. (2018); Oneto et al. (2020); Huynh et al. (2020) studies statistical

learning with optimal transport. Studies like Genevay et al. (2017); Bousquet et al. (2017); Sanjabi et al. (2018); Deshpande et al. (2019); Lei et al. (2019); Patrini et al. (2020); Onken et al. (2021) utilize optimal transport distance as a metric for improving the performance and robustness of unsupervised learning algorithms. The Sinkhorn algorithm, featured in works such as Fernando et al. (2013); Redko et al. (2017); Courty et al. (2017); Alvarez-Melis et al. (2018); Nguyen et al. (2022; 2021); Xu et al. (2022); Turrisi et al. (2022), is commonly used in robust domain adaptation, particularly in addressing distribution shift. Additionally, various approaches employ neural networks to learn the optimal transport map, as seen in (Seguy et al., 2017; Damodaran et al., 2018; Makkuva et al., 2020; Mokrov et al., 2021; Daniels et al., 2021).

## 3 VARIATIONAL FORM OF THE SINKHORN ALGORITHM

This section summarizes the variational form of the Sinkhorn algorithm, a mathematical representation crucial for understanding our proposed algorithm's theoretical underpinnings. As pointed out, the Sinkhorn algorithm performs alternating maximization for the Lyapunov potential. By introducing the Lagrangian variable and using the minimax theorem (for a detailed derivation, see Appendix A), we formulate the associated primal-dual problem to equation 1 as:

$$\max_{x,y} \min_P L(P, x, y) := \frac{1}{\eta} P \cdot \log P + C \cdot P - x \cdot (P\mathbf{1} - r) - y \cdot (P^\top \mathbf{1} - c).$$

The Lyapunov function $f$, as described in equation 2, comes from eliminating $P$ (see Appendix A):

$$f(x, y) = \min_P L(P, x, y).$$

Maximizing over $f$ is equivalent to solving the problem defined in equation 1: As a consequence of the minimax theorem, $(x^\star, y^\star) = \arg\max_{x,y} f(x, y)$ effectively solves equation 1, as the following equation shows:

$$\arg\min_{P:P\mathbf{1}=r, P^\top \mathbf{1}=c} C \cdot P + \frac{1}{\eta} H(P) =: P^\star = \exp\left(\eta(-C + x^\star \mathbf{1}^\top + \mathbf{1}(y^\star)^\top) - 1\right).$$

Let $P$ be defined as $\exp\left(\eta(-C + x\mathbf{1}^\top + \mathbf{1}y^\top) - 1\right)$, where it serves the intermediate matrix corresponding to dual variables $x, y$. As $f$ is concave, the first-order condition is equivalent to optimality. Upon direct calculation, one has

$$\partial_{x_i} f(x, y) = r_i - \sum_k P_{ik}, \quad \partial_{y_j} f(x, y) = c_j - \sum_k P_{kj}.$$

As a consequence, maximizing $x$ with fixed $y$ corresponds to scaling the rows of $P$ so that $P\mathbf{1} = r$. Likewise, maximizing $y$ with fixed $x$ corresponds to scaling the column of matrix $P$ so that $P^\top \mathbf{1} = c$. Thus, the Sinkhorn matrix scaling algorithm corresponds to an alternating coordinate ascent approach to the Lyapunov function, as illustrated in equation 3. For the reader's convenience, we write down the first and second derivatives of the Lyapunov function $f$:

$$\nabla_x f(x, y) = r - P\mathbf{1}, \quad \nabla_y f(x, y) = c - P^\top \mathbf{1}, \quad \nabla^2 f(x, y) = \eta \begin{bmatrix} \text{diag}(P\mathbf{1}) & P \\ P^\top & \text{diag}(P^\top \mathbf{1}) \end{bmatrix} \quad (4)$$

## 4 MAIN ALGORITHM

We introduce Algorithm 1, herein referred to as Sinkhorn-Newton-Sparse (SNS). This algorithm extends the Sinkhorn algorithm with a second stage featuring a Newton-type subroutine. In short, Algorithm 1 starts with running the Sinkhorn algorithm for $N_1$ steps, then switches to a sparsified Newton algorithm for fast convergence. Algorithm 1 employs a straightforward thresholding rule for the sparsification step. Specifically, any entry in the Hessian matrix smaller than a constant $\rho$ is truncated to zero, and the resulting sparsified matrix is stored in a sparse data structure. The truncation procedure preserves symmetry and diagonal dominance of the Hessian matrix (as a simple consequence of equation 4), which justifies the use of conjugate gradient for linear system solving (Golub & Van Loan, 2013). The obtained search direction $\Delta z$ is an approximation to the exact Newton step search direction, i.e., the solution to the linear system $(\nabla^2 f)v = -\nabla f$. In other words, removing sparse approximation will recover the Newton algorithm, and the Newton algorithm without sparsification is considered in Appendix D. In Appendix C, we introduce additional techniques used for improving the numerical stability of SNS.

---

**Algorithm 1** Sinkhorn-Newton-Sparse (SNS)

---

**Require:** $f, x_{\text{init}} \in \mathbb{R}^n, y_{\text{init}} \in \mathbb{R}^n, N_1, N_2, \rho, i = 0$

1: # Sinkhorn stage
2: $(x, y) \leftarrow (x_{\text{init}}, y_{\text{init}})$         ▷ Initialize dual variable
3: **while** $i < N_1$ **do**
4:      $P \leftarrow \exp\left(\eta(-C + x\mathbf{1}^\top + \mathbf{1}y^\top) - 1\right)$
5:      $x \leftarrow x + (\log(r) - \log(P\mathbf{1}))/\eta$
6:      $P \leftarrow \exp\left(\eta(-C + x\mathbf{1}^\top + \mathbf{1}y^\top) - 1\right)$
7:      $y \leftarrow y + \left(\log(c) - \log(P^\top\mathbf{1})\right)/\eta$
8:      $i \leftarrow i + 1$
9: **end while**
10: # Newton stage
11: $z \leftarrow (x, y)$
12: **while** $i < N_1 + N_2$ **do**
13:      $M \leftarrow \text{Sparsify}(\nabla^2 f(z), \rho)$         ▷ Truncate with threshold $\rho$
14:      $\Delta z \leftarrow \text{Conjugate\_Gradient}(M, -\nabla f(z))$    ▷ Solve sparse linear system
15:      $\alpha \leftarrow \text{Line\_search}(f, z, \Delta z)$       ▷ Line search for step size
16:      $z \leftarrow z + \alpha \Delta z$
17:      $i \leftarrow i + 1$
18: **end while**
19: Output dual variables $(x, y) \leftarrow z$.

---

**Complexity analysis of Algorithm 1** Inside the Newton stage, the conjugate gradient method involves $O(n)$ left multiplications by $M := \text{Sparsify}(\nabla^2 f(z), \rho)$. Furthermore, left multiplication by $M$ involves $O(\tau(M)n^2)$ arithmetic operations, where $\tau(\cdot)$ is the sparsity defined in Definition 1. Thus, obtaining $\Delta z$ is of complexity $O(\tau(M)n^3)$. To maintain an upper bound for per-iteration complexity, in practice, one sets a target sparsity $\lambda$ and picks $\rho$ dynamically to be the $\lceil \lambda n^2 \rceil$-largest entry of $\nabla^2 f(z)$, which ensures $\tau(M) \leq \lambda$. For the forthcoming numerical experiments, whenever the transport problem has unique solutions, we have found setting $\lambda = 2/n$ suffices for a convergence performance quite indistinguishable from a full Newton step, which is corroborated by the approximate sparsity results mentioned in Theorem 1.

**Necessity of Sinkhorn steps** We remark that the Sinkhorn stage in SNS has two purposes. The first purpose is warm initialization. Using the Sinkhorn steps, we bring the intermediate dual variable $(x, y)$ closer to the optimizer so that the subsequent Newton step has a good convergence speed. Secondly, this proximity ensures that the intermediate matrix $P$ satisfies approximate sparsity. While the number of Sinkhorn steps is written as a fixed parameter in Algorithm 1, one can alternatively switch to the Newton stage dynamically. In particular, we switch to a sparsified Newton step when the two following conditions hold: First, the intermediate matrix $P$ should admit a good sparse approximation, and secondly, the Newton step should improve the Lyapunov potential more than the Sinkhorn algorithm. Our analysis in Section 5 demonstrates that it requires at most $O(1/\epsilon^2)$ Sinkhorn steps for the sparsification to be within an error of $\epsilon$.

## 5 SPARSITY ANALYSIS OF SNS

This section gives a complete theoretical analysis of the approximate sparsity throughout the SNS algorithm: Theorem 1 analyzes the approximate sparsity of the Hessian after the $N_1$ Sinkhorn step; Theorem 2 analyzes the approximate sparsity within the $N_2$ Newton steps and proves monotonic improvement on the approximate sparsity guarantee. Three symbols are heavily referenced throughout this analysis, which we list out for the reader's convenience:

- $P_{t,\eta}$ : The result of Sinkhorn algorithm after $t$ iterations.

- $\mathcal{F}$ : The set of optimal transport plan in the original problem

- $P_\eta^\star$ : The entropy-regularized optimal solution,

where $\mathcal{F}, P_\eta^\star$ satisfy the following equation:

$$\mathcal{F} := \underset{P:P\mathbf{1}=r,P^\top\mathbf{1}=c}{\arg\min} C \cdot P, \quad P_\eta^\star := \underset{P:P\mathbf{1}=r,P^\top\mathbf{1}=c}{\arg\min} C \cdot P + \frac{1}{\eta}H(P). \tag{5}$$

**Approximate sparsity in the Sinkhorn stage**     We first prove the main theorem on the approximate sparsity of $P_{t,\eta}$, which in particular gives an approximate sparsity bound for the Hessian matrix when one initiates the sparsified Newton step in Algorithm 1. We generalize to allow for the setting where potentially multiple solutions exist to the optimal transport problem. Definition 2 lists the main concepts in the analysis. For concepts such as a polyhedron, face, and vertex solution, the readers can consult Cook et al. (1998) for detailed definitions.

**Definition 2.** *Define $\mathcal{P}$ as the feasible set polyhedron, i.e. $\mathcal{P} := \{P \mid P\mathbf{1} = r, P^\top\mathbf{1} = c, P \geq 0\}$. The symbol $\mathcal{V}$ denotes the set of vertices of $\mathcal{P}$. The symbol $\mathcal{O}$ stands for the set of optimal vertex solution, i.e.*

$$\mathcal{O} := \underset{P\in\mathcal{V}}{\arg\min} C \cdot P. \tag{6}$$

*The symbol $\Delta$ denotes the vertex optimality gap*

$$\Delta = \min_{Q\in\mathcal{V}-\mathcal{O}} Q \cdot C - \min_{P\in\mathcal{O}} P \cdot C.$$

*We use $\mathcal{F} = \mathrm{Conv}(\mathcal{O})$ to denote the optimal face to the optimal transport problem, and $\tau(\mathcal{F}) := \max_{M\in\mathcal{F}} \tau(M)$ is defined to be the sparsity of $\mathcal{F}$. We define a distance function to $\mathcal{F}$ by $\mathrm{dist}(\mathcal{F}, P) = \arg\min_{M\in\mathcal{F}}\|M - P\|_1$.*

We move on to prove the theorem for approximate sparsity of $P_{t,\eta}$:

**Theorem 1.** *Assume $\|r\|_1 = \|c\|_1 = 1$, and let $\Delta$ be as in Definition 2. There exists constant $q, t_1$ such that, for $\eta > \frac{1+2\log n}{\Delta}, t > t_1$, one has*

$$\mathrm{dist}(\mathcal{F}, P_{t,\eta}) \leq 6n^2 \exp\left(-\eta\Delta\right) + \frac{\sqrt{q}}{\sqrt{t}}.$$

*Therefore, for $\lambda^\star = \tau(\mathcal{F})$, it follows that $P_{t,\eta}$ is $(\lambda^\star, \epsilon_{t,\eta})$-sparse, whereby*

$$\epsilon_{t,\eta} := 6n^2 \exp\left(-\eta\Delta\right) + \frac{\sqrt{q}}{\sqrt{t}}.$$

*As a result, the Hessian matrix in Algorithm 1 after $N_1$ Sinkhorn steps is $(\frac{\lambda^\star}{2} + \frac{1}{2n}, 2\epsilon_{N_1,\eta})$-sparse.*

*Define the subset $\mathcal{S} \subset \mathbb{R}^{n\times n}$ so that $C \in \mathcal{S}$ if $C$ is a cost matrix for which $\min_{P:P\mathbf{1}=r,P^\top\mathbf{1}=c} C \cdot P$ has non-unique solutions. Then, $\mathcal{S}$ is of Lebesgue measure zero, and so the optimal transport $\min_{P:P\mathbf{1}=r,P^\top\mathbf{1}=c} C \cdot P$ has unique solution generically. The generic condition $C \notin \mathcal{S}$ leads to $\lambda^\star \leq 2/n$. If one further assumes $r = c = \frac{1}{n}\mathbf{1}$, then $\lambda^\star = 1/n$.*

*Proof.* It suffices to construct a sparse approximation to $P_{t,\eta}$ that matches the requirement in Definition 1. We choose the sparse approximation to be the matrix $P^\star \in F$ which satisfies $P^\star = \arg\min_{M\in F}\|P_\eta^\star - M\|_1$. By triangle inequality, one has

$$\|P_{t,\eta} - P^\star\| \leq \epsilon := \|P_{t,\eta} - P_\eta^\star\|_1 + \mathrm{dist}(\mathcal{F}, P_\eta^\star).$$

Thus $P_{t,\eta}$ is $(\tau(\mathcal{F}), \epsilon)$-sparse. Existence of a sparse approximation to the Hessian matrix $\nabla^2 f$ is shown by the following construction:

$$\tilde{M} = \begin{bmatrix} \mathrm{diag}(P_{t,\eta}\mathbf{1}) & P^\star \\ (P^\star)^\top & \mathrm{diag}(P_{t,\eta}^\top\mathbf{1}) \end{bmatrix}.$$

One can directly count that the number of nonzero entries of $\tilde{M}$ is upper bounded by $2n + 2\lambda^\star n^2$. Moreover, direct computation shows $\|\tilde{M} - \nabla^2 f\|_1 = 2\|P_{t,\eta} - P^\star\|_1 \leq 2\epsilon$. Thus, we show that the Hessian matrix is $(\frac{\lambda^\star}{2} + \frac{1}{2n}, 2\epsilon)$-sparse.

Thus, the proof reduces to proving $\epsilon < \epsilon_{t,\eta}$. By Corollary 9 in (Weed, 2018), if $\eta > \frac{1+2\log n}{\Delta}$, it follows

$$\mathrm{dist}(\mathcal{F}, P_\eta^\star) \leq 2n^2 \exp\left(-\eta\Delta + 1\right) \leq 6n^2 \exp\left(-\eta\Delta\right). \tag{7}$$

By Pinsker inequality (Pinsker, 1964) and Theorem 4.4 in Ghosal & Nutz (2022), there exists constants $q, t_1$ such that, for any $t > t_1$, one has

$$\|P_{t,\eta} - P_\eta^\star\|_1^2 \leq \mathrm{KL}(P_{\eta,t}\|P_\eta^\star) + \mathrm{KL}(P_\eta^\star\|P_{\eta,t}) \leq \frac{q}{t}, \tag{8}$$

and thus $\|P_{t,\eta} - P_\eta^\star\|_1 \leq \frac{\sqrt{q}}{\sqrt{t}}$. Combining equation 7 and equation 8, one has $\epsilon < \epsilon_{t,\eta}$, as desired.

One has $C \in \mathcal{S} \in \mathbb{R}^{n \times n}$ if and only if there exist two distinct permutation matrices $P_1, P_2$ so that $C \cdot P_1 = C \cdot P_2$. For each $P_1, P_2$, the condition that $C \cdot P_1 = C \cdot P_2$ is on a subset of measure zero on $\mathbb{R}^{n \times n}$. As there are only a finite number of choices for $P_1, P_2$, it follows that $\mathcal{S}$ is a finite union of sets of measure zero, and thus in particular $\mathcal{S}$ is of measure zero. Thus, one has $C \notin \mathcal{S}$ generically.

When $\min_{P:P\mathbf{1}=r,P^\top\mathbf{1}=c} C \cdot P$ has a unique solution $\mathcal{P}^\star$, it must be an extremal point of the polyhedron $\mathcal{P}$, which has $2n - 1$ non-zero entries (Peyré et al., 2017), and therefore $\tau(\mathcal{F}) = \tau(\mathcal{P}^\star) \leq \frac{2n-1}{n^2} \leq \frac{2}{n}$. In the case where $r = c = \frac{1}{n}\mathbf{1}$, it follows from Birkhoff–von Neumann theorem that $nP^\star$ is a permutation matrix, and therefore $\tau(\mathcal{F}) = \tau(\mathcal{P}^\star) = \frac{1}{n}$. □

As Theorem 1 shows, taking a target sparsity $\lambda = O(1/n) > 3n/2$ in Algorithm 1 leads to a good sparse approximation, which leads to a $O(n^2)$ per-iteration cost. It is worth pointing out that the $\exp(-\Delta\eta)$ term in $\epsilon_{t,\eta}$ shows the appealing property that the matrix $P_{t,\eta}$ has a better sparse approximation guarantee in the challenging case where $\eta$ is very large.

**Approximate sparsity in the Newton stage** We consider next the approximate sparsity inside the $N_2$ Newton loops. We show that approximate sparsity is monotonically improving as the Newton step converges to the optimal solution:

**Theorem 2.** *Let $z_k = (x_k, y_k)$ denote the dual variable at the $k$-th Newton step, and let $\epsilon_k = \max_z f(z) - f(z_k)$ be the sub-optimality gap for $z_k$. For the sake of normalizing the transport plan formed by $z_k$, take $y_{k,\star} = \arg\max_y f(x_k, y)$ and define $P_k$ to be the transport plan formed after column normalization, i.e.,*

$$P_k = \exp\left(\eta(-C + x_k\mathbf{1}^\top + \mathbf{1}(y_{k,\star})^\top) - 1\right).$$

*Then, for the same constant $q$ in Theorem 1, for $\epsilon_k < 1$ and $\eta > \frac{1+2\log n}{\Delta}$, the matrix $P_k$ is $(\lambda^\star, \zeta_k)$-sparse, where*

$$\zeta_k = 6n^2 \exp(-\eta\Delta) + \sqrt{q}(\epsilon_k)^{1/4}.$$

The proof is similar to that of Theorem 1, and we defer it to Appendix B. We remark that the $\epsilon_k^{1/4}$ term in Theorem 2 becomes insignificant if $\epsilon_k$ decreases at a super-exponential rate. Moreover, the statement in Theorem 2 would have shown monotonically improving approximate sparsity if one were to add a column scaling step inside the Newton inner loop in Algorithm 1. For clarity, we do not include any matrix scaling in the Newton stage of Algorithm 1. However, combining different techniques in the second step might aid convergence and is worth investigating.

**Sparsity under non-uniqueness** We explore sparsity guarantees when the optimal transport problem lacks a unique solution. Although these cases are rare in practice, we point out the somewhat surprising result: There exist conditions for which one can derive sparsity properties of the optimal face using tools from extremal combinatorics on bipartite graphs (Erdős & Spencer, 1974; Jukna, 2011). On a high level, the optimality of $P^\star \in \mathcal{F}$ forbids certain substructures from forming in an associated bipartite graph, which in turn gives an upper bound on the number of nonzero entries of $P^\star$. We defer the proof to Appendix B.

**Theorem 3.** *For a cost matrix $C = [c_{ij}]_{i,j=1}^n \in \mathbb{R}^{n \times n}$, suppose that $c_{ij} + c_{i'j'} = c_{i'j} + c_{ij'}$ if and only if $i = i'$ or $j = j'$. Then one has $\tau(\mathcal{F}) \leq \frac{1+o(1)}{\sqrt{n}}$.*

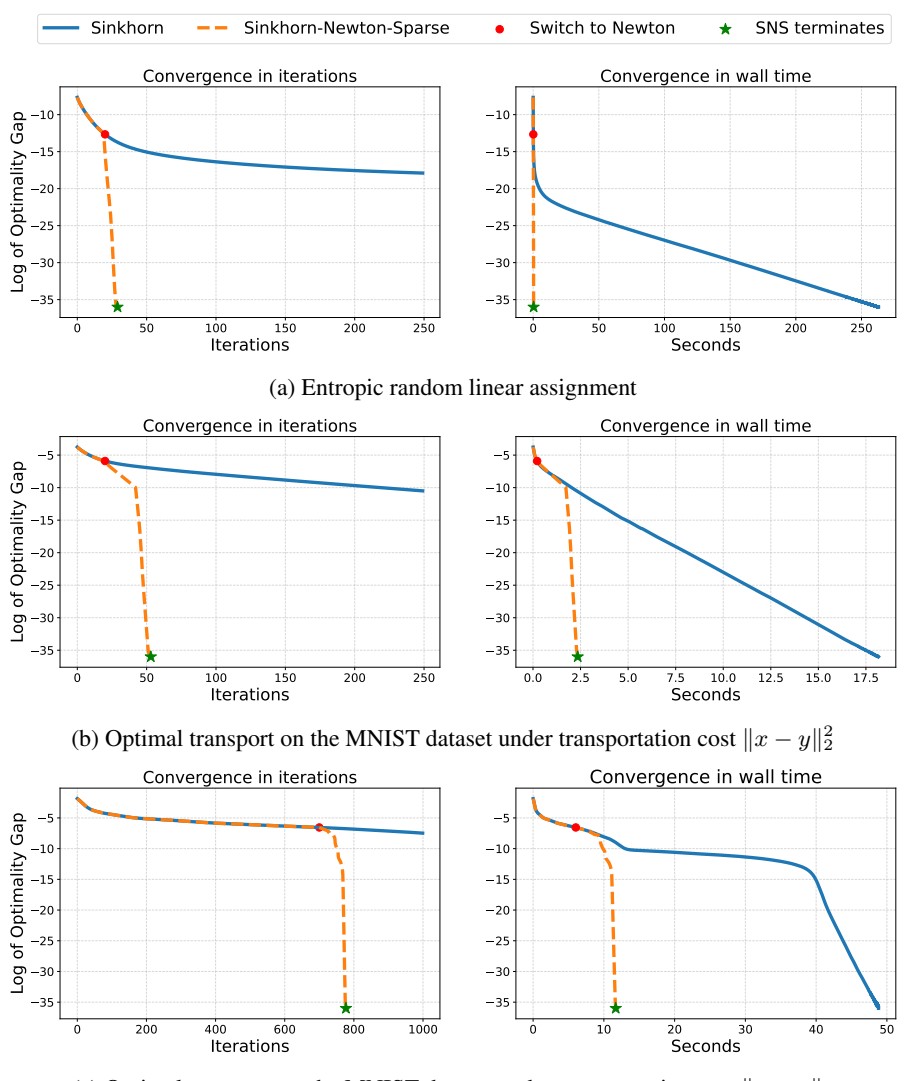

(a) Entropic random linear assignment

(b) Optimal transport on the MNIST dataset under transportation cost $\|x - y\|_2^2$

(c) Optimal transport on the MNIST dataset under transportation cost $\|x - y\|_1$

Figure 1: Performance comparison between Algorithm 1 and the Sinkhorn algorithm.

## 6 NUMERICAL RESULT

We conduct numerical experiments to compare the original Sinkhorn and SNS algorithms. We use the following settings: We obtained the runtime data on a 2021 Macbook Pro laptop with an Apple M1 Pro chip. The linear system solving is done through the conjugate gradient step as mentioned in Algorithm 1. To set a challenging case, we use an entropic regularization with $\eta = 1200$ throughout the experiments. We refer the reader to Appendix F for the performance of SNS under different entropy parameter $\eta$. We defer a comparison of SNS with the Sinkhorn-Newton algorithm (Brauer et al., 2017) to Appendix D, where we show that removing sparsification from SNS results in a prohibitively expensive algorithm due to its $O(n^3)$ runtime complexity. Additionally, we perform experiments on the numerical performance of quasi-Newton methods (Nocedal & Wright, 1999).

In the first numerical test, we consider the random assignment problem with entropic regularization (Mézard & Parisi, 1987; Steele, 1997; Aldous, 2001), considered a hard instance of optimal transport. The cost matrix $C = [c_{ij}]_{ij=1}^n \in \mathbb{R}^{n \times n}$ with $n = 500$ is generated by $c_{ij} \sim \text{Unif}([0, 1])$. The source and target vectors are $c = r = \frac{1}{n}\mathbf{1}$. We run Algorithm 1 with $N_1 = 20$ and a target sparsity of $\lambda = 2/n$. Figure 1a shows that SNS drastically outperforms Sinkhorn in iteration and runtime.

Table 1: Performance comparison between SNS and Sinkhorn. Both algorithms are run until they reach machine accuracy.

| Case | Method | Stage | Time (s) | Iterations |
|---|---|---|---|---|
| Random | SNS | Sinkhorn | 0.12 | 20 |
| | | Newton | 0.22 | 9 |
| | | Total | **0.34** | **29** |
| | Sinkhorn | Total | 233.36 | 56 199 |
| MNIST L2 | SNS | Sinkhorn | 0.17 | 20 |
| | | Newton | 2.16 | 33 |
| | | Total | **2.33** | **53** |
| | Sinkhorn | Total | 18.84 | 2041 |
| MNIST L1 | SNS | Sinkhorn | 6.28 | 700 |
| | | Newton | 5.94 | 77 |
| | | Total | **12.22** | **777** |
| | Sinkhorn | Total | 45.75 | 5748 |

In the second numerical experiment, similar to the experiment setting in Cuturi (2013), we consider the more practical case of optimal transport on the MNIST dataset. In particular, two images are respectively converted to a vector of intensities on the $28 \times 28$ pixel grid, which are then normalized to sum to 1. The entry corresponding to the $(i_1, i_2)$-th pixel is conceptualized as the point $(i_1/28, i_2/28) \in \mathbb{R}^2$, and the transport cost is the Euclidean distance cost $\|x - y\|^2$. Similarly, we pick $N_1 = 20$ in Algorithm 1 with a target sparsity of $\lambda = 2/n$. Figure 1b shows a similar performance boost to the Sinkhorn algorithm.

As the approximate sparsity analysis underlying Section 5 mainly focuses on the situation of unique solutions, it is of interest to test problems with many optimal transport solutions, as it is a case where the SNS algorithm might potentially break in practice. For this purpose, we consider the MNIST example under the $l_1$ transport cost $\|x - y\|_1$, which is known to have non-unique solutions due to the lack of convexity in the $\|\cdot\|_1$ norm (Villani et al., 2009). As the Sinkhorn algorithm converges quite slowly, we pick $N_1 = 700$ before switching to the Newton stage. Choosing a target sparsity of $\lambda = 15/n$ is sufficient for convergence to the ground truth, which shows that SNS runs quite well even without a uniqueness guarantee.

In Table 1, we benchmark the runtime for Sinkhorn versus SNS to reach machine accuracy, which shows that SNS has an overall runtime advantage for convergence. In particular, we list the performance of SNS in the Newton stage, which shows that early stopping of Sinkhorn matrix scaling steps and switching to the Newton stage results in an order of magnitude speedup in iteration counts. While we cannot prove the conjectured super-exponential convergence, the low iteration count in the Newton stage shows strong numerical support.

## 7 CONCLUSION

We propose the Sinkhorn-Newton-Sparse algorithm, demonstrating its empirical super-exponential convergence at a $O(n^2)$ per-iteration cost through numerical validation. We prove several novel bounds on approximate sparsity underlying the algorithm. For problems with non-unique solutions, we elucidate a novel relationship between approximate sparsity and extremal combinatorics. We contend that this new method significantly advances the field of high-precision computational optimal transport and complements the existing Sinkhorn algorithm.

For future work, it may be interesting to study how the approximation accuracy of the sparsification step affects the algorithm's convergence and how to devise more sophisticated sparse approximation techniques beyond simple thresholding. Moreover, it is an exciting direction to incorporate existing optimal transport techniques with the SNS algorithm, including greedy row/column optimization, dynamic regularization scheduling, and hardware-based parallel accelerations. Theoretically, analytic properties on the Lyapunov potential might provide more insight into the region where the sparsified Newton algorithm achieves the super-exponential convergence we empirically observe.

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

## A  EQUIVALENCE OF PRIMAL AND PRIMAL-DUAL FORM

We now show that the primal form in equation 1 can be obtained from the primal-dual form by eliminating the dual variables.

**Proposition 1.** *Define*

$$L(P, x, y) = \frac{1}{\eta} P \cdot \log P + C \cdot P - x \cdot (P\mathbf{1} - r) - y \cdot (P^\top \mathbf{1} - c),$$

*and then the following equation holds:*

$$\max_{x,y} \min_P L(P, x, y) = \min_{P : P\mathbf{1} = r, P^\top \mathbf{1} = c} \frac{1}{\eta} P \cdot \log P + C \cdot P \tag{9}$$

*Moreover, for the Lyapunov potential function $f$ in equation 2, one has*

$$f(x, y) = \min_P L(P, x, y). \tag{10}$$

*Proof.* The use of Lagrange multiplier implies the following equality:

$$\min_P \max_{x,y} L(P, x, y) = \min_{P : P\mathbf{1} = r, P^\top \mathbf{1} = c} \frac{1}{\eta} P \cdot \log P + C \cdot P.$$

As $L$ is concave in $x, y$ and convex in $P$, one can invoke the minimax theorem to interchange the operations of maximization and minimization. Therefore,

$$\max_{x,y} \min_P L(P, x, y) = \min_{P : P\mathbf{1} = r, P^\top \mathbf{1} = c} \frac{1}{\eta} P \cdot \log P + C \cdot P \tag{11}$$

In terms of entries, one writes $L(P, x, y)$ as follows:

$$\max_{x_i, y_j} \min_{p_{ij}} \frac{1}{\eta} \sum_{ij} p_{ij} \log p_{ij} + \sum_{ij} c_{ij} p_{ij} - \sum_i x_i (\sum_j p_{ij} - r_i) - \sum_j y_j (\sum_i p_{ij} - c_j). \tag{12}$$

We then solve the inner min problem explicitly by taking the derivative of $p_{ij}$ to zero, from which one obtains

$$p_{ij} = \exp(\eta(-c_{ij} + x_i + y_j) - 1).$$

Plugging in the formula for $p_{ij}$, one has

$$\min_P L(P, x, y) = -\frac{1}{\eta} \sum_{ij} \exp(\eta(-c_{ij} + x_i + y_j) - 1) + \sum_i r_i x_i + \sum_j c_j y_j = f(x, y).$$

$\square$

## B  PROOF OF THEOREM 2 AND THEOREM 3

We present the proof as follows:

*Proof.* (of Theorem 2) Same as Theorem 1, we use the proof strategy that the approximate sparsity guarantee monotonically improves as $P_k$ converges to $P_\eta^\star$. By Lemma 3.2 in Ghosal & Nutz (2022) and Pinsker's inequality, for $\alpha_k = \mathrm{KL}(r \| P_k \mathbf{1})$, one has

$$\|P_k - P_\eta^\star\|_1^2 \leq \mathrm{KL}(P_k \| P_\eta^\star) + \mathrm{KL}(P_\eta^\star \| P_k) \leq q \min(\alpha_k, \sqrt{\alpha_k}).$$

By Lemma 2 in Altschuler et al. (2017), one has

$$\alpha_k \leq \max_z f(z) - f(x_k, y_{k,\star}) \leq \max_z f(z) - f(x_k, y_k) = \epsilon_k,$$

where the second inequality comes from the definition of $y_{k,\star}$. From the proof in Theorem 1, there exists $P_\star \in \mathcal{F}$ such that

$$\|P_k - P^\star\|_1 \leq 6n^2 \exp(-\eta \Delta) + \sqrt{q} \min\left((\epsilon_k)^{1/2}, (\epsilon_k)^{1/4}\right).$$

Therefore, the statement in the theorem specializes to the more relevant case where $\epsilon_k < 1$. $\square$

*Proof.* (of Theorem 3)

We begin by constructing the bipartite graph associated with an optimal transport plan $P^\star$. Let $A, B$ be two index sets with $A \cap B = \emptyset$ and $|A| = |B| = n$. For an optimal transport plan $P^\star \in \mathbb{R}^{n \times n}$, we define its associated bipartite graph $G_{P^\star} = (A \cup B, E_{P^\star})$, where $(i, j) \in E_{P^\star}$ if and only if the $(i, j)$-th entry of $P^\star$ is non-zero.

Suppose that there exists $i, i' \in A$ and $j, j' \in B$ with $i \neq i'$, $j \neq j'$, such that $P^\star$ is nonzero on the $(i, j), (i, j'), (i', j), (i', j')$ entries. Then, one can consider a perturbation $E$ to $P^\star$. Specifically, $E$ is $+1$ on entries $(i, j), (i', j')$, and is $-1$ on entries $(i, j'), (i', j)$, and is zero everywhere else. As $E\mathbf{1} = E^\top \mathbf{1} = \mathbf{0}$, for sufficiently small $\epsilon$, it follows that $P^\star \pm \epsilon E$ is still feasible.

We note that one must have $C \cdot E = 0$, otherwise one would contradict the optimality of $P^\star$, but this would mean $c_{ij} + c_{i'j'} = c_{ij'} + c_{i'j}$, which contradicts the assumption on $C$. Thus, $P^\star$ cannot be simultaneously nonzero on the $(i, j), (i, j'), (i', j), (i', j')$ entries. By the definition of $E_{P^\star}$, it would mean that $G_{P^\star}$ cannot simultaneously contain the edges $(i, j), (i, j'), (i', j), (i', j')$. Thus, in terms of graph-theoretic properties, we have shown that $G_{P^\star}$ is $C_{2,2}$ free, whereby $C_{2,2}$ is the $2 \times 2$ clique. Then, by the $C_{2,2}$ free property in Theorem 2.10 in (Jukna, 2011) shows $E_{P^\star}$ cannot have more than $n\sqrt{n} + n$ edges. Thus $\tau(P^\star) \leq \frac{1 + o(1)}{\sqrt{n}}$, as desired. $\qquad\square$

## C   SINKHORN-NEWTON-SPARSE FOR AUGMENTED LYAPUNOV FUNCTION

Since the Lyapunov function $f$ satisfies $f(x, y) = f(x + \gamma\mathbf{1}, y - \gamma\mathbf{1})$ for any scalar $\gamma$, this implies that $f$ has a degenerate direction of $v = \begin{bmatrix} \mathbf{1} \\ -\mathbf{1} \end{bmatrix}$. Thus, to maintain numerical stability, we use in practice an augmented Lyapunov potential

$$f_{\mathrm{aug}}(x, y) := f(x, y) - \frac{1}{2}(\sum_i x_i - \sum_j y_j)^2.$$

Switching to the augmented Lyapunov potential does not change the task, as $(x^\star, y^\star) = \arg\max_{x,y} f_{\mathrm{aug}}(x, y)$ is simply the unique maximizer of $f$ which satisfies $\sum_i x_i^\star = \sum_j y_j^\star$. Moreover, as $\nabla^2 f_{\mathrm{aug}} = \nabla^2 f - vv^\top$, one can adapt the Hessian approximation to a superposition of rank-1 and sparse matrix (Candès et al., 2011), which will likewise lead to $O(n^2)$ complexity.

We introduce Algorithm 2, a variation of Algorithm 1. This altered version uses the augmented Lyapunov potential $f_{\mathrm{aug}}$ to accommodate the degenerate direction $v = \begin{bmatrix} \mathbf{1} \\ -\mathbf{1} \end{bmatrix}$:

$$f_{\mathrm{aug}}(x, y) := f(x, y) - \frac{1}{2}(\sum_i x_i - \sum_j y_j)^2.$$

As the discussion in Section 4 shows, optimizing for $f_{\mathrm{aug}}$ is equivalent to optimizing for $f$. Algorithm 2 differs from the original algorithm in two respects. First, the initialization of $z$ is through the output of the Sinkhorn stage after projection into the orthogonal complement of $v$. Second, in the Newton stage, one obtains the Hessian approximation term $M$ with the superposition of a sparse matrix $\mathrm{Sparsify}(\nabla^2 f(z), \rho)$ and a rank-1 matrix $vv^\top$. Importantly, for $\lambda = \tau(\mathrm{Sparsify}(\nabla^2 f(z), \rho))$, the cost of left multiplication of $M$ is $O(\lambda n^2) + O(n)$. As the $O(n)$ term is dominated by the $O(\lambda n^2)$ term, the conjugate gradient step still has $O(\lambda n^3)$ scaling. Overall, the computational complexity of Algorithm 2 is nearly identical to Algorithm 1.

---

**Algorithm 2** Sinkhorn-Newton-Sparse (SNS) with augmented Lyapunov potential

---

**Require:** $f_{\mathrm{aug}}, x_{\mathrm{init}} \in \mathbb{R}^n, y_{\mathrm{init}} \in \mathbb{R}^n, N_1, N_2, \rho, i = 0$

1: # Sinkhorn stage

2: $v \leftarrow \begin{bmatrix} \mathbf{1} \\ -\mathbf{1} \end{bmatrix}$  ▷ Initialize degenerate direction

3: $(x, y) \leftarrow (x_{\mathrm{init}}, y_{\mathrm{init}})$  ▷ Initialize dual variable

4: **while** $i < N_1$ **do**

5:      $P \leftarrow \exp\left(\eta(-C + x\mathbf{1}^\top + \mathbf{1}y^\top) - 1\right)$

6:      $x \leftarrow x + (\log(r) - \log(P\mathbf{1}))/\eta$

7:      $P \leftarrow \exp\left(\eta(-C + x\mathbf{1}^\top + \mathbf{1}y^\top) - 1\right)$

8:      $y \leftarrow y + \left(\log(c) - \log(P^\top\mathbf{1})\right)/\eta$

9:      $i \leftarrow i + 1$

10: **end while**

11: # Newton stage

12: $z \leftarrow \mathrm{Proj}_{v^\perp}((x, y))$  ▷ Project into non-degenerate direction of $f$

13: **while** $i < N_1 + N_2$ **do**

14:      $M \leftarrow \mathrm{Sparsify}(\nabla^2 f(z), \rho) - vv^\top$  ▷ Truncate with threshold $\rho$.

15:      $\Delta z \leftarrow \mathrm{Conjugate\_Gradient}(M, -\nabla f_{\mathrm{aug}}(z))$  ▷ Solve sparse linear system

16:      $\alpha \leftarrow \mathrm{Line\_search}(f_{\mathrm{aug}}, z, \Delta z)$  ▷ Line search for step size

17:      $z \leftarrow z + \alpha \Delta z$

18:      $i \leftarrow i + 1$

19: **end while**

20: Output dual variables $(x, y) \leftarrow z$.

---

## D  SINKHORN-NEWTON WITHOUT SPARSITY

In this section, we show that the Sinkhorn-Newton algorithm (Brauer et al., 2017) without accounting for Hessian sparsity would be prohibitively slower than the SNS Algorithm. We remark that the Sinkhorn-Newton algorithm can be obtained from SNS by removing the Sparsify step in Algorithm 1. In this case, one solves for the descent direction by directly using the Hessian, i.e., changing to

$$\Delta z = -\left(\nabla^2 f(z)\right)^{-1} \nabla f(z).$$

Theoretically, this leads to a $O(n^3)$ per-iteration complexity, which is considerably costlier than our best-scenario complexity of $O(n^2)$ under $O(1/n)$ sparsity.

To empirically verify the impracticality of this method, we repeat the entropic random linear assignment problem experiment in Section 6 with $n = 2000$, $N_1 = 20$ and $\eta = 5000$. Table 2 summarizes our findings. As expected, we observe that the Sinkhorn-Newton method is significantly slower than SNS, especially in terms of per-iteration complexity. For larger $n$, the Sinkhorn-Newton algorithm will be even more unfavorable.

Table 2: Performance comparison between SNS and Sinkhorn-Newton during the Newton stage. Both algorithms are run until they reach machine accuracy.

| Method | Time (s) | Iterations | Time per iteration (s) |
|---|---|---|---|
| SNS | **3.26** | 11 | **0.30** |
| Sinkhorn-Newton | 118.56 | **10** | 11.86 |

## E  COMPARISON BETWEEN SINKHORN-NEWTON-SPARSE WITH QUASI-NEWTON METHODS

This section presents the result of quasi-Newton algorithms (Nocedal & Wright, 1999) applied to entropic optimal transport problems. We show that, while being a reasonable proposal for solving entropic optimal transport with second-order information, traditional quasi-Newton algorithms

work poorly in practice. In short, a Quasi-Newton algorithm can be obtained from SNS by replacing the Hessian approximation step in Algorithm 1. Specifically, instead of sparse approximation, a quasi-Newton method approximates the Hessian $M$ through the history of gradient information. In particular, we consider the Broyden–Fletcher–Goldfarb–Shanno (BFGS) algorithm and the limited-memory Broyden–Fletcher–Goldfarb–Shanno (L-BFGS) algorithm, which are two of the most widely used quasi-Newton methods.

We repeat the three experiment settings in Section 6, and the result is shown in Figure 2. To ensure a fair comparison, the quasi-Newton candidate algorithms are given the same Sinkhorn initialization as in the SNS algorithm. As the plot shows, quasi-Newton algorithms do not show significant improvement over the Sinkhorn algorithm. Moreover, in the two experiments based on the MNIST dataset, both quasi-Newton candidates perform worse than the Sinkhorn algorithm in terms of runtime efficiency. As the iteration complexity of the quasi-Newton candidates does not exhibit the numerical super-exponential convergence shown in SNS, we conclude that noisy Hessian estimation from gradient history accumulation is inferior to direct sparse approximation on the true Hessian matrix.

# F  SINKHORN-NEWTON-SPARSE UNDER DIFFERENT ENTROPY REGULARIZATION PARAMETER

In this section, we show an acceleration of SNS over the Sinkhorn algorithm under a wider range for the entropy regularization parameter $\eta$. In particular, we focus on the setting of MNIST image under $l_1$ and $l_2$ costs. Importantly, it is common practice to pixel distance is used to form the cost matrix. For example, the Earth-Mover distance (EMD) considered in (Altschuler et al., 2017) is defined by

$$d_{\text{pixel}}\left((i,j),(i',j')\right) := |i - i'| + |j - j'|.$$

In our work, a pixel $(i,j)$ is embedded to the point $(i/28, j/28) \in \mathbb{R}^2$ before the distance function is applied. Thus, this text uses the normalized distance function

$$d\left((i,j),(i',j')\right) := \left|\frac{i-i'}{28}\right| + \left|\frac{j-j'}{28}\right|.$$

As $d = \frac{1}{28}d_{\text{pixel}}$, our choice of $\eta = 1200$ in Section 6 is equivalent to choosing $\eta = 1200/28 \approx 42$. As the range used in Altschuler et al. (2017) is $\eta \in [1, 9]$, our $\eta$ is similar to the range of entropy regularization commonly used. To show the performance of SNS is robust under different $\eta$, we benchmark the performance of SNS under an extended practical choice of $\eta = 28k$ for $k = \{1, 3, 5, 7, 9, 11\}$. For the Sinkhorn stage warm initialization, we take $N_1 = 10k + 100$ for each $\eta = 28k$, and the target sparsity is taken to be $\lambda = 15/n$. In Table 3, we show the performance of SNS compared with the Sinkhorn algorithm to reach machine accuracy. One can see that the SNS algorithm consistently outperforms the Sinkhorn algorithm, and the improvement is more significant under larger choices of $\eta$.

For further validation, in Figure 3a we plot the Wasserstein $W_1$ transport distance for the entropy regularized optimal solution $P_\eta^\star$ under different $\eta$, which shows indeed that $\eta \approx 150$ is sufficient for the practical goal of obtaining transport plan with relatively low transport cost.

For the case of squared $l_2$ distance, as the squared $l_2$ distance is scaled by a factor of 576, we benchmark the performance of SNS under $\eta = 576k$ for $k = \{1, 3, 5, 7, 9, 11\}$. As can be seen in Figure 3, one needs $\eta \approx 1000$ to reach within $1\%$ accuracy of the ground-truth transport cost, which is why the range of $\eta$ considered is a reasonable choice. For the Sinkhorn stage warm initialization, we take $N_1 = 10k$ for each $\eta = 576k$, and the target sparsity is taken to be $\lambda = 4/n$. In Table 4, we show the performance of SNS compared with the Sinkhorn algorithm to reach machine accuracy. One can see that the SNS algorithm consistently outperforms the Sinkhorn algorithm, and the improvement is more significant under larger choices of $\eta$. Moreover, for the case of $\eta = 576$, even though the presence of entropy regularization is strong, the numerical result shows that $\lambda = 4/n$ in the sparse Hessian approximation is sufficient to reach machine accuracy rapidly.

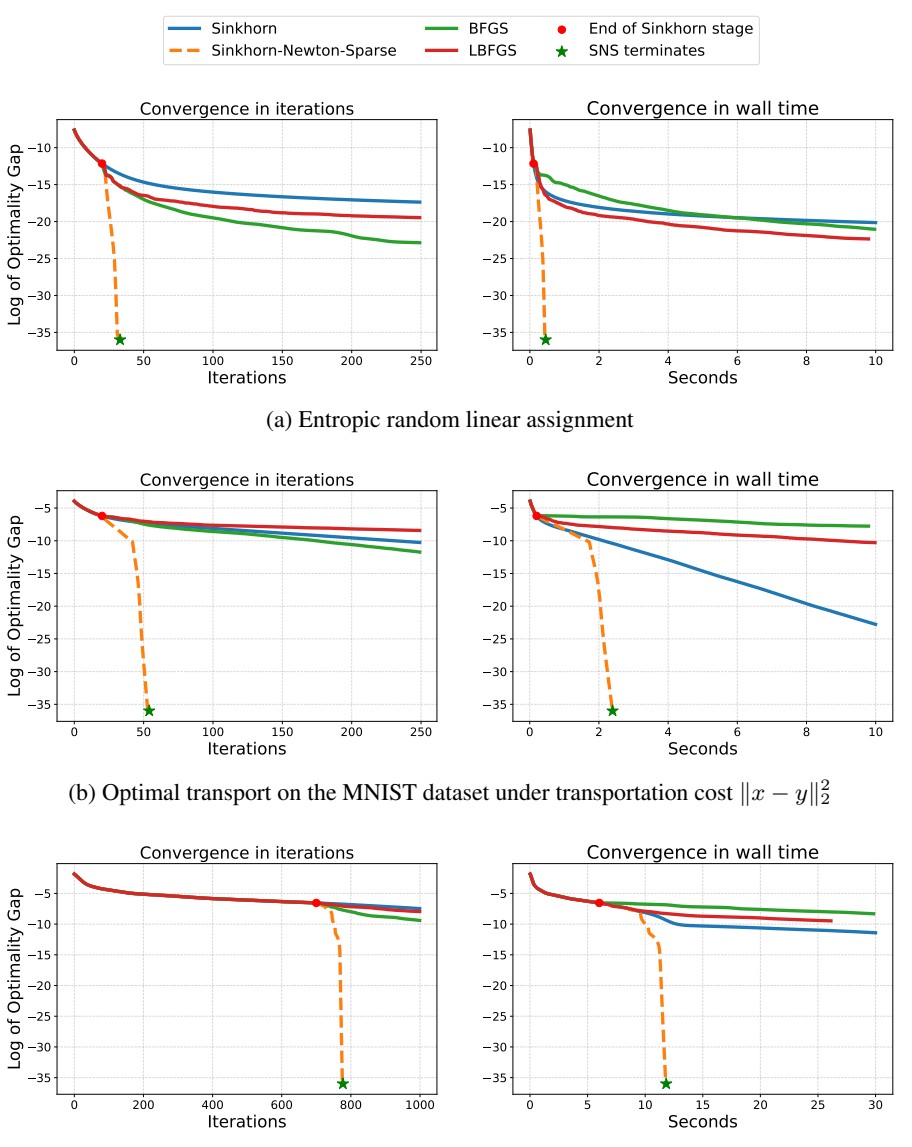

(a) Entropic random linear assignment

(b) Optimal transport on the MNIST dataset under transportation cost $\|x - y\|_2^2$

(c) Optimal transport on the MNIST dataset under transportation cost $\|x - y\|_1$

Figure 2: Performance of Quasi-Newton methods, compared against the Sinkhorn-Newton-Sparse algorithm and the Sinkhorn algorithm.

Table 3: Performance comparison between SNS and Sinkhorn for different $\eta$ under the transportation cost $\|x-y\|_1$. Both algorithms are run until they reach machine accuracy. The time and iteration of the SNS method refers to the combined time and iteration of the two stages combined.

| Entropy | Method | Time (s) | Iterations |
|---|---|---|---|
| $\eta = 28$ | SNS | 1.45 | 110 |
| | Sinkhorn | 1.96 | 173 |
| $\eta = 84$ | SNS | 5.32 | 147 |
| | Sinkhorn | 9.63 | 899 |
| $\eta = 140$ | SNS | 5.41 | 167 |
| | Sinkhorn | 14.53 | 1399 |
| $\eta = 196$ | SNS | 6.10 | 189 |
| | Sinkhorn | 15.56 | 1499 |
| $\eta = 252$ | SNS | 7.78 | 216 |
| | Sinkhorn | 17.81 | 1699 |
| $\eta = 308$ | SNS | 8.08 | 236 |
| | Sinkhorn | 19.76 | 1899 |

Table 4: Performance comparison between SNS and Sinkhorn for different $\eta$ under the transportation cost $\|x-y\|_2^2$. Both algorithms are run until they reach machine accuracy. The time and iteration of the SNS method refers to the combined time and iteration of the two stages combined.

| Entropy | Method | Time (s) | Iterations |
|---|---|---|---|
| $\eta = 576$ | SNS | 3.40 | 33 |
| | Sinkhorn | 9.95 | 946 |
| $\eta = 1728$ | SNS | 5.10 | 64 |
| | Sinkhorn | 32.92 | 3072 |
| $\eta = 2880$ | SNS | 7.08 | 96 |
| | Sinkhorn | 62.96 | 6083 |
| $\eta = 4032$ | SNS | 9.70 | 134 |
| | Sinkhorn | 108.24 | 10 596 |
| $\eta = 5184$ | SNS | 12.83 | 177 |
| | Sinkhorn | 166.97 | 16 299 |
| $\eta = 6336$ | SNS | 22.30 | 259 |
| | Sinkhorn | 248.43 | 23 498 |

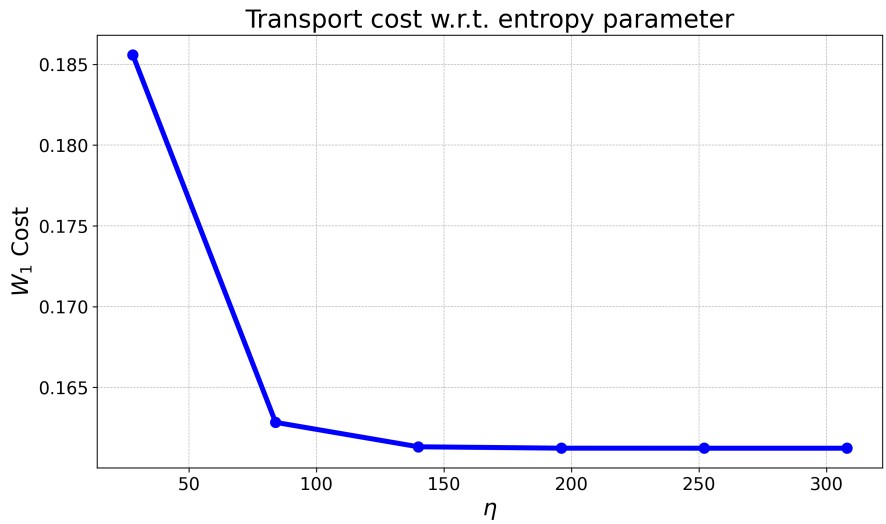

(a) MNIST dataset under transportation cost $\|x - y\|_1$

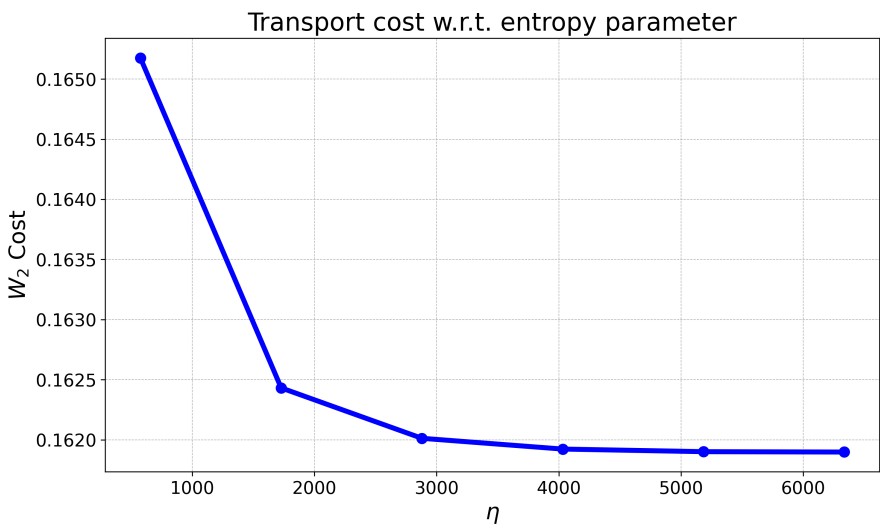

(b) MNIST dataset under transportation cost $\|x - y\|_2^2$

Figure 3: Optimal transport cost of the obtained entropic regularized solution for different $\eta$.

