# OpenReview forum: "Accelerating Sinkhorn algorithm with sparse Newton iterations"
_ICLR.cc/2024/Conference — ICLR 2024 poster_

### Official Review · Reviewer_orpT · 2023-10-30

**Soundness:** 3 good
**Presentation:** 3 good
**Contribution:** 3 good
**Rating:** 8
**Confidence:** 3

**Summary:**

The authors focus on accelerating the Sinkhorn algorithm for optimal transport (OT). The Sinkhorn algorithm is an iterative matrix scaling algorithm used to solve the entropy-reularized OT problem. Its per-iteration complexity is $O(n^2)$ for discrete distributions of size $n$. Its convergence rate behaves like a polynomial converging method, which can be slow. Other optimization methods like second-order methods enjoy super-exponential convergence, but a greater per-itereration cost $O(n^3)$ (e.g. Newton). Here they find a link between the Sinkhorn iterations and the sparsity of the Hessian of the Lyapunov potential. Once the Hessian is sparse, the per-iteration cost can reach $O(n^2)$, hence the same per-iteration cost as Sinkhron while benefiting from the faster convergence rate of the second-order methods. The authors propose to combine the two methods, i.e. Newton iteration once the Hessian is sparse, which improves the convergence rate. They provide quantitative results to validate the algorithm.

**Strengths:**

- The overall presentation of the manuscript is good, and the method is clear.
- The method accelerates the convergence of the Sinkhorn algorithm without increasing the per-iteration cost.
- The method is theoretically motivated by theorem 1; we know that after some Sinkhorn steps the Hessian can be sparse, hence the per-iteration cost of Newton's methods is lower.

**Weaknesses:**

- I find the evaluation a bit limited. In particular, it would be great to see more results for different regularization parameter, and include standard deviation in the tables.
- It would be great to have comparison with other methods than Sinkhorn, although I agree that most methods can be combined with SNS, it would be beneficial to compare their convergence.
- Have you tried removing the Sinkhorn stage  ($N_1=0$) ? I would like to understand how much of the sparsity is due to Sinkhorn (Thm1) vs the `Sparsify` step in Alg.1, and its influence on the convergence.

**Questions:**

Questions and minor comments.

- I believe that to compute the optimality gap (e.g. Fig.1) you need a ground truth (i.e. the minimizer), in this case how is it computed ? Is it the OT plan without entropy regularization ?
- I might be missing something, but I don't see the target sparsity $\lambda$ in Alg.1.
- How do you choose the number of Sinkhorn steps $N_1$? Would it be possible to switch to the Newton stage once the Hessian is sparse enough ?
- The related work section is great, but maybe you can add other references that speed up Sinkhorn using a factorization of the ground cost [1,3] or Chebyshev polynomials with a sparse graph [2].
- The results in Tab.1 are on the number of iterations until reaching machine accuracy. They don't inform us on the quality of the approximation. It could be interesting to compare the accuracy of the plan in the case where we have a closed form (e.g. entropic OT between Gaussian distributions [4]).
- The authors could add a brief review of Newton’s algorithm since it is an important piece of the paper. In particular, on the importance of solving the Hessian system.
- In eq. 1, you need to define $\eta$ and specify if it should be greater than zero.
- Having a preliminary or background section could help. Parts of the introduction could be in this section, and definition such as the optimality gap could be added in that section.
- In section 2 "Convergence of Sinkhorn", what is $\alpha$ ? Should it be $\eta$ ?

[1] Scetbon, Meyer, Marco Cuturi, and Gabriel Peyré. "Low-rank Sinkhorn factorization." International Conference on Machine Learning. PMLR, 2021.

[2] G. Huguet, A. Tong, M. R. Zapatero, C. J. Tape, G. Wolf and S. Krishnaswamy, "Geodesic Sinkhorn For Fast and Accurate Optimal Transport on Manifolds," 2023 IEEE 33rd International Workshop on Machine Learning for Signal Processing (MLSP).

[3] Scetbon, Meyer, and Marco Cuturi. "Linear time Sinkhorn divergences using positive features." Advances in Neural Information Processing Systems 33 (2020): 13468-13480.

[4] Mallasto, A., Gerolin, A., & Minh, H. Q. (2022). Entropy-regularized 2-Wasserstein distance between Gaussian measures. Information Geometry, 5(1), 289-323.

---

> ### Author Response · Authors · 2023-11-18
> **Response to main concerns**
>
> ### Response to the performance evaluation of SNS
> **Regularization parameter**
>
> The reviewer suggests that it would be beneficial to try different regularization parameters. We thank the reviewer for the helpful suggestion, which greatly improves the clarity of this work. To systematically address the question,
> - We added experiment for L1 and L2 MNIST examples with a broader range of practical $\eta$ in Appendix F.
> - We show that we are able to achieve systematic performance improvement similar to the result shown in the main experiment in Section 6.
>
> **Clarification on contribution**
> - In our work, we have considered optimal transport in three different examples. As the main application of optimal transport is usually the approximating the $W_1$, $W_2$ distances, our L1 and L2 examples are typical instances of entropic optimal transport in geometric problems. In high-dimensional transport problems, such as Domain Adaptation using OT (e.g., Fernando et al., 2013; Redko et al., 2017),  the formed cost matrix $C$ will be of high rank, which is why we use a randomly generated matrix $C$ to reflect this type of situation. We believe that the three examples cover a rather wide range of practical applications within optimal transport.
> - We clarify that the performance improvement in our work is consistent across all tested instances, showing minimal variation dependent on the instance. Therefore, we simplified the presentation by presenting the work over one instance, even though the conclusion would hold if the comparison were run over many instances.
>
> ### Response on Sinkhorn stage
>
> Regarding sparsity from the Sinkhorn algorithm, its primary role comes in the form of providing a warm initialization to enter a super-exponential convergence stage. In practical cases, even if the costly Sinkhorn-Newton algorithm (Brauer et al. (2017)) were used, the improvement on the iteration count over SNS will be limited. What this means is that SNS does a very good job of capturing the Hessian structure of the problem. However, if the initial point is not near the minimum, then second-order method won't be able to enter the region where super-exponential convergence can be achieved. Thus, a good practice would be to keep $N_1 \neq 0$. **We will provide further analysis on the effect of $N_1$ and post our result before the end of the review period.**
>
> The reviewer also comments that we can potentially combine the sparse Newton stage with other warm initializations. The reviewer is correct, and indeed this is a very immediate and potentially fruitful future research direction following our work. The authors chose to use Sinkhorn algorithm as the initialization because Sinkhorn-based warm initialization is the only method with a proven exponential convergence guarantee. As first-order methods will not have exponential convergence, the only warm initialization strategy that might have an exponential convergence guarantee will be GreekHorn (Altschuler et al, 2017), but such a guarantee has not been proven to the best of our knowledge. Thus, we choose the Sinkhorn initialization despite that it may be slower than other methods in practice. As mentioned in the conclusion section, a combination of sparse newton with other warm initialization is easily implementable for future research, and might lead to even better performance on entropic optimal transport.

---

> ### Author Response · Authors · 2023-11-18
> **Response to questions**
>
> In what follows, we address the reviewer's concerns one by one.
>
> > Q1: I believe that to compute the optimality gap (e.g. Fig.1) you need a ground truth (i.e. the minimizer), in this case how is it computed? Is it the OT plan without entropy regularization?
>
> A1: As Sinkhorn algorithm can be thought of as a matrix scaling task, our termination criterion is when the rows and columns are scaled to machine accuracy. The minimizer is the dual variable which gives rise to the entropic optimal transport solution with the target $\eta$. (Detailed information on how dual variable connects with the entropic transport plan can be found in Section 2 of our paper)
>
> > Q2: I might be missing something, but I don't see the target sparsity $\lambda$ in Alg.1.
>
> A2: The reviewer's observation is accurate; target sparsity is indeed not specified in Alg.1. In general, one can truncate a matrix $M \in \mathbb{R}^{n \times n}$ by setting the truncation term $\rho$ to be the $k$-th largest entry of the Hessian, so that one would achieve a $k/n^2$ target sparsity. This point is also made in Section 4 under paragraph "Complexity analysis of Algorithm 1".
>
> > Q3: How do you choose the number of Sinkhorn steps $N_1$? Would it be possible to switch to the Newton stage once the Hessian is sparse enough?
>
> A3: Right now we usually select $N_1 = O(1)$ for warm initialization. A better choice is to use a dynamic approach. For example, one can perform one Newton step every twenty Sinkhorn step, and see if one achieves significant improvement over the Lyapunov potential in that one Newton step. Once Newton step improves the Lyapunov potential much better than Sinkhorn step, then one would switch fully to Newton stage. We opted not to include this in the text to maintain a clear focus on the main technical novelty of our work. Approximate sparsity can also serve as a metric for switching, though the transport plan is usually sparse in practice after a few steps of Sinkhorn.
>
> > Q4: The related work section is great, but maybe you can add other references that speed up Sinkhorn using a factorization of the ground cost [1,3] or Chebyshev polynomials with a sparse graph [2].
>
> A4: The authors thank the reviewer for the suggestion. We have incorporated these references into the related works section.
>
> > Q5: The results in Tab.1 are on the number of iterations until reaching machine accuracy. They don't inform us on the quality of the approximation. It could be interesting to compare the accuracy of the plan in the case where we have a closed form (e.g. entropic OT between Gaussian distributions [4]).
>
> A5: The authors thank the reviewer for the suggestion.  In Appendix F, we include results on how entorpic transport solution converges in terms of transport cost for MNIST L1 and L2 examples (see Figure 3 under the revised text). In addition, we thank the reviewer for bringing the work [4] to our attention. We have included this work in the related literature section.
>
> > Q6: The authors could add a brief review of Newton’s algorithm since it is an important piece of the paper. In particular, on the importance of solving the Hessian system.
>
> A6: We thank the reviewer for the suggestion. Due to space constraints, a detailed review of Newton's algorithm is omitted, but we direct readers to key references for comprehensive understanding, including Convex optimization by Boyd \& Vandenberghe and Numerical Optimization by Wright \& Nocedal, which are excellent references. We have added a comment in the main text that removing sparsification will lead to the Newton's algorithm for the Sinkhorn objective. We also added a comment on the importance of solving the Hessian in achieving super-exponential convergence.
>
> > Q7: In eq. 1, you need to define $\eta$ and specify if it should be greater than zero.
>
> A7: The authors thank the reviewer for the suggestion. We will edit the manuscript accordingly.
>
> > Q8: Having a preliminary or background section could help. Parts of the introduction could be in this section, and definition such as the optimality gap could be added in that section.
>
> A8: We thank the reviewer for the suggestion. We tried to change the organization of the paper but didn't find a viable way to reorganize the text.
>
> > Q9: In section 2 "Convergence of Sinkhorn", what is $\alpha$ ? Should it be $\eta$ ?
>
> A9: The definition of $\alpha$ is in footnote 2 of the submitted manuscript prior to edit. To improve clarity, we have included the definition of $\alpha$ in the paragraph directly.
>
> **We sincerely hope that we have answered your concerns, in which case, we would greatly appreciate it very much if you could increase your scores. Please also reach out to us in the discussion phase if there are additional questions**

---

> > ### Comment · Reviewer_orpT · 2023-11-21
> >
> > I would like to thank the authors for their response and for addressing my questions.
> >
> > There are still two points I would like to mention:
> >
> > - In Fig.3 of the appendix, could you also show the results for Sinkhorn (on the same figure)? Just to show that the two methods converge to a similar transport plan.
> > - The sparsify step in Alg.1 (line 13) might cause an approximation error. It would be good to quantify this error or to show the difference between distances computed with or without sparsification. From appendix D Tab.2, we know that it is much faster with sparsification, but it would strengthen the results to show that the distances or the plans are similar.

---

> ### Author Response · Authors · 2023-11-21
>
> Dear Reviewer orpT,
>
> We are happy to address the question, and please let us know if you have further questions.
>
> > In Fig.3 of the appendix, could you also show the results for Sinkhorn (on the same figure)? Just to show that the two methods converge to a similar transport plan.
>
> Figure 3 shows how the the transport cost of optimal transport plan $P_{\eta}^{\star}$ converges with respect to $\eta$, and therefore it doesn't have a direct link to SNS or Sinkhorn.
>
> Specifically, in this case, we are plotting the transport cost for the optimal transport plan $P_{\eta}^{\star}$ obtained from the entropic optimal transport problem. Both Sinkhorn and SNS are algorithms to find a transport plan to converge to the point $P_{\eta}^{\star}$. As we plot the transport plan *P_{\eta}^{\star}* which both algorithms converge to, the figure we plotted here is agnostic to the algorithm.
>
> > The sparsify step in Alg.1 (line 13) might cause an approximation error. It would be good to quantify this error or to show the difference between distances computed with or without sparsification. From appendix D Tab.2, we know that it is much faster with sparsification, but it would strengthen the results to show that the distances or the plans are similar.
>
> In this work, we mainly are concerned with the efficiency for Sinkhorn or any other method to reach the entropic optimal solution $P_{\eta}^{\star}$. Therefore, as both SNS and Sinkhorn-Newton converge to the optimal solution $P_{\eta}^{\star}$ in a few steps, we can conclude that both methods give rise to the same transport cost.
>
> We emphasize that our method uses sparsification in the optimization subroutine, but we do not change the optimization task itself, and therefore our method, once converged, will have the same optimal solution as Sinkhorn's algorithm or Sinkhorn-Newton.
>
> **We greatly appreciate the reviewer's question! This gives us a chance to clarify our methodology. If you have any other questions, we will be glad to answer them.**
>
> (Update to reply: Some further explanation on Figure 3 and detailed answer on closeness in distance and transport plan)
>
> As the result in Section 6 shows, both SNS and Sinkhorn algorithms are capable of converging to the ground truth transport plan $P_{\eta}^{\star}$ with machine accuracy. Thus, the plot has the interpretation of the cost of the transport plan for both SNS and Sinkhorn when they converge to machine accuracy.
>
> We shall go through a more direct and quantitative response. For the convergence of SNS and Sinkhorn to the optimal transport plan, the proof of Theorem 2 in Appendix B shows that the total variation distance with respect to $P_{\eta}^{\star}$ is (up to a constant) bounded by the square root of the optimality gap. Therefore, the convergence plot in Figure 1 shows that both Sinkhorn and SNS can reach 10^{-8} accuracy in terms of total variation distance with respect to the optimal transport plan $P_{\eta}^{\star}$. As the cost matrix $C$ (defined in Section 1) is $O(1)$ entry-wise, if we were to plot the transport cost of SNS and Sinkhorn at the convergence point, it will likewise lead to a ~10^{-8} difference in the $W_1$ cost obtained by $P_{\eta}^{\star}$, or a ~10^{-4} difference in the $W_2$ cost. The authors judged the difference to be too negligible to put in the plot.

---

> > ### Comment · Reviewer_orpT · 2023-11-21
> >
> > Thank you for the clarification, it is clearer for me now. I don't have any more questions at the moment.

---

> > > ### Author Response · Authors · 2023-11-21
> > >
> > > Thank you for your valuable feedback and active participation! Please let us know if there’s anything more we can do to increase the score. Additionally, we would be delighted to receive any suggestions you might have to refine our manuscript.

---

### Official Review · Reviewer_CUDR · 2023-10-31

**Soundness:** 3 good
**Presentation:** 4 excellent
**Contribution:** 3 good
**Rating:** 8
**Confidence:** 4

**Summary:**

This paper proposes a method for the fast numerical solution of the Kantorovich problem with entropic regularization. The submission is closely related to the cited work of Brauer, Clason, Lorenz and Wirth that explored the use of Newton iterations for this problem. The main contributions are two-fold:
1. The observation (that the authors prove) that after several Sinkhorn iterations, the Hessian matrix becomes close to sparse.
2. An algorithm based on this observation that combines several iterations of Sinkhorn followed by Newton iterations with a sparsified Hessian. The Hessian is sparsified by zeroing-out small entries.

**Strengths:**

* The paper focuses on the important and timely problem of efficiently computing regularized optimal transport.
* The authors clearly explain the potential advantages of second-order methods while noting the inefficiency of classic Newton iterations. This motivates their approach of using a sparse approximation to the Hessian.
* The convergence of the Hessian to a nearly-sparse matrix throughout the steps of the algorithm is proved (in both the Sinkhorn and Newton stages).
* The method is clearly explained and the results should be easy to reproduce.

**Weaknesses:**

* A potential red flag is that, in the numerical results, only a single entropic regularization coefficient is considered (\eta=1200). I suspect that this is a weak regularization term which results in rather sparse couplings and is therefore favorable to their method. It is OK if the method is not beneficial in the strong-regularization regime. However, this needs to be acknowledged in the paper and not swept under the rug.

* The method is not compared to any other current methods aside from the classical methods on which it is based (Sinkhorn and Newton).

These two weaknesses taken together are the main reason for my "marginally below" score. If they are properly addressed I will increase my rating of the paper.

**Questions:**

* For the numerical section, I would like to see numerical results for the same data sets with several regularization strengths (e.g. \eta=10,100,1000). Additionally, I think it is important to compare the proposed method to other current methods for accelerating OT with entropic regularization. In particular to "Massively scalable Sinkhorn distances via the Nystrom method" by Altschuler, Bach, Rudi, and Niles-Weed (2019).

* Have you considered any quasi-Newton methods? Will these be applicable in the context of your paper?

---

> ### Author Response · Authors · 2023-11-18
>
> ### Response on the choice of entropy parameter $\eta$
>
> The reviewer comments that the entropy regularization term $\eta$ is too large, therefore the regularization effect is too weak. We appreciate the very detailed feedback and acknowledge this concern. To address it, we added Appendix F with experiments with a wide range of $\eta$. Additionally, we would like to motivate the original choice of the value of the regularization parameter.
>
> The manuscript maps the (i,j)-th pixel to the point $(i/28, j/28)$, and then use L1 or L2 distance, which respectively corresponds to the Earth mover distance (EMD) and the $W_2$ distance. As a result, for the calculation of EMD (see the L1 distance example in Section 6), our cost matrix C used is 28 times smaller than using pixel distance. To align with studies using pixel distances, our $\eta = 1200$ effectively equates to $\eta \approx 42$ after accounting for the scaling factor. The choice of $\eta \approx 42$ is a reasonable choice in the OT research community. For instance, (Altschuler et al., 2017) uses $\eta = 1,5, 9$ for a 28*28 grid. For the L2 distance, we have not found a choice of $\eta$ among recent work, but we found that taking $\eta \approx 1000$ in the L2 case enables estimation of $W_2$ distance with roughly an 1\% error, which is similar to the desired level of accuracy in estimating EMD in (Altschuler et al., 2017).
>
> In Appendix F, we systematically address the question with experiments for L1 and L2 MNIST examples with a wider range of practical $\eta$. The SNS algorithm achieves performance improvement similar to the result shown in the main experiment in Section 6. As such, our algorithm shows a significant performance boost in very common instances of OT. If Appendix F and the reasoning behind the original choice address the reviewer's concern, we kindly ask the reviewer to reconsider the review score.
>
> ### Response on Quasi-Newton method
>
> The reviewer comments that we should compare our work more with existing methods beyond Sinkhorn and Newton's method. We include in Appendix E the numerical comparison between our algorithm with quasi-Newton methods, which shows that quasi-Newton method performs much worse than our proposed method. Moreover, we cannot observe significant improvement over even the original Sinkhorn algorithm.
>
> ### Response on comparing performance with (Altschuler et al, 2019)
>
> We greatly agree with the reviewer in that (Altschuler et al, 2019) is a technically very impressive work and constitutes one of the major developments in recent OT algorithm research. However, it is important to our work that kernel approximation of $e^{\eta C}$ of any kind is not used. The work in (Altschuler et al, 2019) achieves speedup in the Sinkhorn task by in part sacrificing the rank of the approximated kernel, and therefore there exists a tradeoff in computational speed and accuracy. In our work, one can conceptualize SNS as introducing an approximated preconditioner, but we do not change the optimization task itself. Thus it will be unfair to compare (Altschuler et al, 2019) to our work, though we can comment that their work and SNS respectively achieve order(s) of magnitude speedup when compared to the Sinkhorn algorithm.
>
> ### Aside: Analysis of quasi-Newton's suboptimal performance
>
> The numerical results in Appendix E provide ample evidence that quasi-Newton shows undesirable performance. Furthermore, we reason theoretically why quasi-Newton method is unfavorable in the Sinkhorn setting. The reasons for quasi-Newton working worse than SNS is twofold.
>
> - Firstly, the use of a quasi-Newton method is only valid if the Hessian of the matrix is hard to obtain, and therefore an approximation of the Hessian is used in place of the true Hessian. In our case, obtaining the Hessian is only of $O(n^2)$ cost, and therefore the cost is dominated by solving the linear system involving the Hessian. Therefore, using a direct approximation of the Hessian has a much better accuracy than keeping track of a Hessian approximation such as in the BFGS algorithm.
>
> - Secondly, for a quasi-Newton algorithm such as BFGS, one would often be storing a full dense hessian, which leads to a $O(n^3)$ complexity. If a limited memory version of quasi-Newton algorithm is used, then the lack of hessian approximation accuracy hampers fast convergence. Thus, a dense Hessian approximation in a typical quasi-Newton algorithm (e.g. BFGS) is guaranteed to have a worse performance than Sinkhorn-Newton, while a limited memory Hessian approximation (e.g. in LBFGS) also tends to have a worse performance than SNS.
>
> **We hope that we have answered the reviewer's concern (especially on the role of entropy regularization parameter), in which case, we would greatly appreciate if the scores could be increased to reflect it. Please also feel free to reach out to us in the discussion phase should you have additional questions.**

---

> > ### Comment · Reviewer_CUDR · 2023-11-22
> > **Happy with this revision**
> >
> > The authors have addressed my concerns with new experiments. Indeed, the main runtime speedup is obtained in the weak regularization regime. However, the proposed method provides speedups across a wide range of regularization strengths.
> >
> > I believe that this, together with the quasi-Newton comparison, strengthens the paper and have upgraded my scores to reflect that.
> >
> > I've noticed one minor typo in Appendix E: "which are the two of the most" which should be "which are two of the most". I'd recommend that the authors run the entire paper through a spellchecker just in case.

---

> ### Comment · Area_Chair_AmX3 · 2023-11-22
> **Replies to author comments**
>
> Dear reviewer,
>
> Thank you very much for your work evaluating this review.
>
> It is critical that you urgently address the author's responses, acknowledge their response, and eventually adjust your rating if warranted.
>
> Best,
>
> AC

---

### Official Review · Reviewer_39Qf · 2023-11-01

**Soundness:** 3 good
**Presentation:** 3 good
**Contribution:** 2 fair
**Rating:** 6
**Confidence:** 3

**Summary:**

This paper introduces the sinkhorn-newton-sparse algorithm to solve the regularised optimal transport problem as described in eq (1). This improves on the previous work by Brauer et al. to the case of sparse transport matrices. The authors define the notion of sparsity as in Definition 1 and provide the rate of convergence of the modified sinkhorn algorithm in theorem 1. The authors then provide the approximate sparsity rate for the newton step in theorem 2. The paper is closed with numerical examples.

**Strengths:**

There is a notable reduction in the number of iterations and the time to convergence while solving a problem using Algorithm 1. It also seems that the definition of sparsity plays a key role in the analysis.

**Weaknesses:**

One notable omission I felt was the lack of guarantees for computational complexity. Although the algorithm performs well numerically, there is no theoretical backing for that.

**Questions:**

Can the authors explain the notion of sparsity introduced in this paper? It seems to me that $\tau$ is just the proportion of non-zero elements. However, is it usual to approximate $M$ by a sparse $\bar M$?

---

> ### Author Response · Authors · 2023-11-18
>
> ### Clarification regarding computational complexity
> The reviewer comments that the algorithm lacks guarantees for computational complexity.
> - We would like to point out that we have a detailed analysis of the per-iteration complexity analysis of Algorithm 1 in Section 4 (paragraph Complexity analysis of Algorithm 1). Our analysis demonstrates that the per-iteration complexity of the Sinkhorn-Newton-Sparse (SNS) algorithm is $O(n^2)$, matching that of the Sinkhorn algorithm.
> - As for iteration complexity, we would like to comment that the Newton stage of SNS in practice convergences with significantly fewer (up to 1000-fold less, as can be seen in the numerical experiment) iterations than the Sinkhorn algorithm. As for theoretical analysis on the iteration count, the authors are constrained by the fact that proving strong convergence in convex optimization algorithm carries significant theoretical challenges, which is a fundamental constraint in the tools available to the convex optimization community. In fact, the main theoretical bottleneck for lack of tools to bound the iteration count of the Sinkhorn-Newton algorithm.
>
> ### Clarification on approximate sparsity
> **More on approximate sparsity**
> As the reviewer pointed out, $\tau$ is just the proportion of non-zero elements for a sparse matrix. For the Hessian matrix M, we use a novel concept of approximate sparsity (see Definition 1). Sparse approximation is a well-discussed topic in applied mathematics, and has been considered in OT community (see Section 2 under paragraph Sparsification in Sinkhorn). An approximately sparse matrix predominantly contains numerically insignificant entries, except for a few significant entries that one needs to keep track of. A $(\lambda, \epsilon)$ approximate sparsity intuitively means that only a $\lambda$ fraction of the entries of the matrix has significant entries, and the combined significance of the rest of the entries is at most $\epsilon$.
>
> **Connection with existing OT work leveraging sparsity**
> Existing works mainly concern about a sparse approximation of the entry-wise exponential of the cost matrix $\exp{(-\eta C)}$, and our work concerns the approximate sparsity of the transport plan matrix $P = X\exp{(-\eta C)}Y$ (see first paragraph of Section 1 for Definition). Moreover, our work is the first to combine sparse approximation with second-order method in the field of OT. Through this choice, we are able to achieve quite order(s)-of-manitude speed-up in matrix scaling. Moreover, to the best of our knowledge, our analysis of sparse approximation in OT in Theorem 1 and Theorem 2 is the first systematic study of approximate sparsity in the entropic regularized transport plan.
>
> **We sincerely hope that we have answered your concerns, in which case, we would greatly appreciate it very much if you could increase your scores. Please also reach out to us in the discussion phase if there are additional questions.**

---

> > ### Comment · Reviewer_39Qf · 2023-12-04
> > **Thank you for the response**
> >
> > Dear Authors,
> >
> > Thank you for the response. I was initially confused about the notion of approximate sparsity introduced; especially because the definition seemed to do a lot of legwork in the theory. However, I see now that this is a special case of the traditional definition of sparsity (in the Frobenius norm sense, which is traditionally used in the literature). This leaves me still somewhat sceptical. Combining the responses to all the other reviewers, I believe that I will keep my original rating.

---

> ### Comment · Area_Chair_AmX3 · 2023-11-22
> **Replies to author comments**
>
> Dear reviewer,
>
> Thank you very much for your work evaluating this review.
>
> It is critical that you urgently address the author's responses, acknowledge their response, and eventually adjust your rating if warranted.
>
> Best,
>
> AC

---

### Official Review · Reviewer_ZDaz · 2023-11-04

**Soundness:** 2 fair
**Presentation:** 2 fair
**Contribution:** 3 good
**Rating:** 6
**Confidence:** 3

**Summary:**

Despite the success of the Sinkhorn algorithm, its runtime may still be slow due to the potentially large number of iterations needed for convergence. To achieve possibly super-exponential convergence, we introduce Sinkhorn-Newton-Sparse (SNS), an extension to the Sinkhorn algorithm, by introducing early stopping for the matrix scaling steps and a second stage featuring a Newton-type subroutine.

**Strengths:**

Sparsification of the Hessian results in a fast per-iteration complexity, the same as the Sinkhorn algorithm. In terms of total iteration count, we observe that the SNS algorithm converges orders of magnitude faster across a wide range of practical cases, including optimal transportation between empirical distributions and calculating the Wasserstein distance of discretized continuous densities. The empirical performance is corroborated by a rigorous bound on the approximate sparsity of the Hessian matrix.

**Weaknesses:**

1. The theoretical results in this paper seem to be simple corollaries of existing resluts.
For example, Eq. (7) which is crucial to the proof of Theorem 1, follows Corollary 9 of (Weed, 2018).
2. Theorem 1 requires that $\eta$ is large enough. However, I believe that a large $\eta$ will bring something negative.
Otherwise, why don't people minimize the original problem without the regularization? Thus, I think the author should remark the condition that $\eta$ is large enough of Theorem 1 and clarify how the value of $\eta$ affect the convergence rate.
3. The paper claims that ``to achieve possibly super-exponential convergence, we introduce Sinkhorn-Newton-Sparse (SNS)''. However, this paper does not provide any sound convergence analysis of SNS.

**Questions:**

No

---

> ### Author Response · Authors · 2023-11-18
> **Response regarding Theorem 1 and entropy regularization**
>
> ### Clarification on the contribution of Theoerem 1
> In this manuscript, the main technical contribution is the acceleration of the existing Sinkhorn algorithm procedure, and the theoretical proof justifies the sparsity argument. For Theorem 1, we synthesize results in (Weed, 2018) (Ghosal \& Nutz, 2022) and (Peyre et al., 2017) into one unified theorem, and we are able to tightly bound the approximate sparsity of the Sinkhorn algorithm after t Sinkhorn steps. Theoerem 1 directly justifies the direction of this paper, which is to use a sparse approximation in the use of the second-order algorithm. This has been corroborated with the numerical experiments, which show a 10-100 fold speedup in compute time. Our main theoretical contribution to Theorem 1 is in the novel synthesis of the presented results. To the best of our knowledge, our work constitutes the very first complete analysis of approximate sparsity of the Hessian matrix of the Lyapunov function (see Section 3, equation (4) for the Hessian, and see equation (2) for the definition of the Lyapunov function). Prior to our analysis, the understanding of this phenomenon was limited, highlighting the novelty of our contribution.
>
> We address the reviewer's comment more directly in terms of novelty. In short, there are two main topics concerning the convergence of entropic optimal transport. First is (a) convergence of the Sinkhorn algorithm to the optimal entropic-regularized solution. Second is (b) convergence of entropic optimal transport to the original linear program solution. The topics (a) and (b) are in their own right fundamental research questions in applied mathematics. The work in (Ghosal \& Nutz, 2022) and (Weed, 2018) are the best analysis in topics (a) and (b) respectively. As shown in (Weed, 2018), its convergence analysis in (b) is shown to have a matching lower bound (Section 3, (Weed, 2018)). Similarly, the analysis in (Ghosal \& Nutz, 2022) is the current strongest result in the exponential convergence of the Sinkhorn algorithm. The authors of this manuscript are successful in providing a tight analysis in approximate sparsity. It is highly unlikely to obtain a stronger bound than the one obtained in this manuscript unless we make substantial improvements in the analysis of either (a) or (b), which would constitute a standalone work on its own.
>
> ### Response on the entropy regularization $\eta$
>
> The reviewer is also correct in pointing out that a large $\eta$ will be bad for the Sinkhorn algorithm. Indeed, in practice, the Sinkhorn algorithm will suffer from choosing a very large $\eta$, which will slow down the algorithm considerably to a runtime that is unfavorable when compared to using a linear programming solver. However, in the case of large $\eta$, a variant of the Sinkhorn algorithm with annealing-based techniques, such as in (Chen et al., 2023), will lead to much better scaling in terms of $\eta$.
>
> In this work, we mainly consider the practical choice of $\eta$. For practical cases of $\eta$, it is usually much faster to use the Sinkhorn algorithm to approximate the optimal transport plan than to run the linear programming code. In Appendix F, we give a detailed discussion to show that our choice of $\eta$ in the experiment is similar to the range of entropy regularization chosen by practitioners. Moreover, we show that a bigger $\eta$ than the one we have chosen will not lead to a significant change in the quality of the solution in terms of the transport cost. For reference, in the L2 MNIST example we tested, our choice of parameter $\eta = 1200$ leads to a mere 1\% difference in transport cost from the ground-truth optimal transport cost.
>
> In addition, one reason for using strong entropy regularization (small $\eta$) is that the transport cost received can be noisy, and entropic regularization can help with dealing with noise, and therefore keeping a small $\eta$ term might help with denoising, as can be seen in the use of entropic regularization in machine learning training, which will follow a similar mechanism as in the paper ``Entropy-SGD: biasing gradient descent into wide valleys".

---

> ### Author Response · Authors · 2023-11-18
> **Clarification on super-exponential convergence**
>
> The reviewer comments that the paper lacks a convergence analysis of SNS. While the point is valid, we would like to point out that the authors are constrained by the fact that proving strong convergence in convex optimization algorithm carries significant theoretical challenges, which is a fundamental constraint in the tools available to the convex optimization community. Conventional analysis in the convergence in convex optimization requires assumptions in the Lipchitz-ness on the utility function f and its Hessian (Boyd \& Vandenberghe, 2004), which is unfortunately unbounded in our case. Moreover, our utility function f, while being concave, does not satisfy self-concordance. Consequently, providing a substantive convergence guarantee for Newton's method in the Sinkhorn objective function is challenging, even without considering sparsity approximation. As such, we do not claim we can prove theoretical super-exponential convergence. Moreover, to address the reviewer's concern, we shall make it more explicit in the manuscript that we cannot prove theoretical super-exponential convergence.
>
> We emphasize that our SNS algorithm has a very strong computational speedup in all of the presented cases. As OT usually deals with geometric transport problems or entropic assignment problems, we have shown strong evidence of speedup across a wide use case of OT. This is also true of Newton's algorithm in general, in the sense that the wide empirical success of Newton's algorithm leads to its wide use by the community even for problems for which it is not clear that the convergence guarantee exists. The same can be said for Sinkhorn's algorithm, for which exponential convergence has only been proven rather recently, but its adoption is due to its good numerical performance and ease of use. In Appendix F, we show that our speedup is robust for different entropy parameters.
>
> Therefore, we emphasize that numerical soundness, aided with a theoretical guarantee for approximate sparsity, lends very strong support for SNS as the future standard of computing OT distance in the machine learning community.
>
> **We sincerely hope that we have answered your concerns, in which case, we would greatly appreciate it very much if you could increase your scores. Please also reach out to us in the discussion phase if there are additional questions.**

---

> ### Comment · Area_Chair_AmX3 · 2023-11-22
> **Replies to author comments**
>
> Dear reviewer,
>
> Thank you very much for your work evaluating this review.
>
> It is critical that you urgently address the author's responses, acknowledge their response, and eventually adjust your rating if warranted.
>
> Best,
>
> AC

---

### Author Response · Authors · 2023-11-18
**General Response to the Reviewers**

We would like to thank the reviewers for their insightful comments and suggestions. We have carefully considered each point and have made several improvements to our manuscript accordingly. In particular,

- **Additional experiments**. We have added an ablation study on the regularization strengths $\eta$ in Appendix F, showing that our proposed algorithm is able to achieve systematic performance improvement similar to the result shown in the main experiment in Section 6. We also include in Appendix E the numerical comparison between our algorithm with quasi-Newton methods, which shows that quasi-Newton method performs much worse than our proposed method.


**Clarifications on our contributions**

 + While our theoretical results are mainly based on some existing works, our theoretical contributions are novel and non-trivial. To the best of our knowledge, our work constitutes the very first complete analysis of the approximate sparsity of the Hessian matrix of the Lyapunov function (see Section 3, equation (4) for the Hessian, and see equation (2) for the definition of the Lyapunov function). To the best of our knowledge, little is understood about the approximate sparsity of the Hessian matrix previous to our analysis.

+ Although our algorithm demonstrates empirical convergence that resembles super-exponential behavior, we have not asserted the ability to establish super-exponential convergence theoretically. We have pointed out in the conclusion section that a more refined theoretical analysis of SNS will be of great important in the future. The usual analysis
in the convergence in convex optimization requires assumption in the Lipchitz-ness on the utility function $f$ and its Hessian (Boyd \& Vandenberghe, 2004)), which is unfortunately unbounded in our case. Moreover, our utility function $f$, while being concave, does not satisfy self-concordance. Therefore, even without sparsity approximation, it would be hard to provide non-vacuous convergence guarantee to Newton’s method in the Sinkhorn objective function.

---

### Author Response · Authors · 2023-11-21
**Request to Review Rebuttal Responses**

Dear Reviewers,

As the discussion phase closes soon, we kindly ask you to review our responses and share any remaining concerns or questions. Thank you for your expertise and dedication.

---

### Meta-Review · Area_Chair_AmX3 · 2023-12-03

**Metareview:**

The reviewers agree that this is a very good submission and that it should be accepted. I encourage the authors to take into account the valuable comments from the reviewers in their camera-ready version.

**Justification For Why Not Higher Score:**

Based on other papers that I have seen at ICLR, both submissions this year and previous years, I think that this paper should be accepted, but that it does not raise to the level of a spotlight/oral, mostly in terms of novelty.

**Justification For Why Not Lower Score:**

I have very much enjoyed this submission, and found it to be very interesting. The reviewers all agree that it should be accepted.

---

### Decision · Program_Chairs · 2024-01-16

Accept (poster)